# Valgarður: A Database of the Petrophysical, Mineralogical, and Chemical Properties of Icelandic Rocks

Samuel W. Scott[1,2], Léa Lévy[3,4], Cari Covell[1], Hjalti Franzson[3], Benoit Gibert[5], Ágúst Valfells[1], Juliet Newson[1], Julia Frolova[6], Egill Júlíusson[7], María Sigríður Guðjónsdóttir[1]

[1]Department of Engineering, Reykjavik University, Reykjavik, 101, Iceland
[2]Institute of Earth Sciences, University of Iceland, Reykjavik, 101, Iceland
[3]Iceland Geosurvey, Reykjavik, 101, Iceland
[4]Faculty of Engineering, Lund University, Lund, 223 63, Sweden
[5]Department of Geosciences, University of Montpellier, Montpellier, 34095, France
[6]Faculty of Geology, Lomonosov Moscow State University, Moscow, Russia
[7]Landsvirkjun, Reykjavik, 103, Iceland

*Correspondence to*: Samuel Scott (samuelwarrenscott@gmail.com)

**Abstract.** The *Valgarður* database is a compilation of data describing the physical and geochemical properties of Icelandic rocks. The dataset comprises 1166 samples obtained from fossil and active geothermal systems, as well as relatively fresh volcanic rocks erupted in sub-aerial or sub-aqueous environments. The database includes petrophysical properties (connected and total porosity, grain density, permeability, electrical resistivity, acoustic velocities, rock strength, thermal conductivity), as well as mineralogical and geochemical data obtained by point-counting, X-ray Fluorescence (XRF), quantitative X-ray
Diffraction (XRD), and Cation Exchange Capacity (CEC) analyses. The database may be accessed at https://doi.org/10.5281/zenodo.6980231 (Scott et al., 2022a). We present the database and use it to characterize the relationship between lithology, alteration, and petrophysical properties. The motivation behind this database is threefold: (i) aid in the interpretation of geophysical data including uncertainty estimations, (ii) facilitate the parameterization of numerical reservoir models, and (iii) improve the understanding of the relationship between rock type, hydrothermal alteration and petrophysical
properties.

## 1 Introduction

The physical properties of igneous and volcanic rocks exert a first-order control on a wide range of geological processes. Rock properties such as porosity and permeability reflect magmatic degassing, eruptive conditions, and environmental conditions related to tectonics, alteration, exhumation and weathering (Petford, 2003; Ceryan et al., 2008; Pola et al., 2012, 2014; Schön,
2015; Colombier et al., 2017; Villeneuve et al., 2019). Variability in the distribution of pore space, fractures, and minerals strongly influences the susceptibility of rock to undergo hydrothermal alteration, which can produce strong changes in mechanical and physical properties (Browne, 1978; Thompson, 1997; Saripalli et al., 2001; Dobson et al., 2003; Cox, 2005;

Franzson et al., 2008; Frolova et al., 2014; Siratovich et al., 2014; Wyering et al., 2014; Sanchez-Alfaro et al., 2016; Heap et al., 2017a; Mordensky et al., 2018; Cant et al., 2018; Navelot et al., 2018; Heap et al., 2020a; Nicolas et al., 2020; Heap et al., 2022a; Weydt et al., 2022). Due to the natural heterogeneity in rock properties, constraining the quantitative relationships between different petrophysical properties and inferring the underlying causes of variability may require extensive petrophysical and mineralogical databases amenable to statistical analysis (Aladejare and Wang, 2017; Asem and Gardoni, 2021). In recent years, a growing number of databases providing detailed petrophysical (Bär et al., 2020; Weinert et al. 2021; Weydt et al., 2021; Heap and Violay, 2021) and geochemical (Gard et al., 2019; Cole et al., 2022; Harðardóttir et al., 2022) data have been published. Such data can be used to build understanding of volcanic eruption risks (Heap et al., 2015; Heap et al., 2021a; Heap et al., 2022b), geothermal resources (Siratovich et al., 2014; Heap et al., 2017b; Heap et al., 2020a; Scott et al., 2022a), surface deformation (Heap et al., 2020b), silicate weathering (Cole et al., 2022) and seismicity (Heap et al., 2015; Meller and Ledéseret, 2021; Heap and Violay, 2021; Heap et al., 2022b).

Basalt is the most common rock type exposed on the surface of the Earth if the area of the ocean floor is included. Owing to the high reactivity of basaltic rocks during surface weathering and water-rock interaction (Gíslason and Eugster, 1987; Stefánsson and Gíslason, 2001; Wolff-Boenisch et al., 2006), basalt plays a major role in the global carbon cycle (Dessert et al., 2003). Accordingly, basaltic rocks are the main target rocks for carbon sequestration efforts involving natural mineral carbonation (Snæbjörnsdóttir et al., 2020). However, compared to sedimentary rocks, which constitute the major source rocks for fossil fuels, and granitic rocks, which comprise much of the continental crust, the petrophysical properties of basaltic rocks are less well-characterized (Heap and Violay, 2021).

Iceland, which is dominantly composed of basalt because of its location astride the Mid-Atlantic Ridge, hosts a large number (>30) of active volcanic systems and associated geothermal systems (Arnórsson, 1995). With continued spreading of the mid-ocean ridge, volcanic systems migrate out of the zone of active volcanism and undergo exhumation and erosion (Walker, 1963; Böðvarsson and Walker, 1964; Pálmason, 1980), exposing altered rocks and intrusive heat sources of so-called 'fossil' geothermal systems at the surface (Friðleifsson, 1983, 1984; Burchardt and Gudmundsson, 2009; Liotta et al., 2020). As a result of hydrothermal alteration at elevated temperatures, these rocks may show complete replacement of primary minerals by secondary alteration minerals (e.g., Franzson et al., 2008).

Iceland's geology has been intensively studied. However, publicly-accessible datasets that provide petrophysical, geochemical, and petrographic data for a given sample set and additionally describe field relations are rare. Studies performed by Orkustofnun (the National Energy Authority) and Iceland Geosurvey (ÍSOR) between 1970-2010 resulted in an extensive dataset consisting of approximately 500 samples analyzed for total and connected porosity, permeability, chemical composition, and petrographic characteristics, which was first released in the *Valgarður*[1] database (Orkustofnun, 2018). This

---

[1]The database is named after Dr. Valgarður Stefánsson (1939-2006), a reservoir physicist who was at the forefront of geothermal exploration in Iceland throughout his career at Orkustofnun. His main geothermal research objective was to define geothermal systems in terms of reservoir characteristics. Recognizing that a relative lack of petrophysical data hampered reliable reservoir modelling, he instigated and headed a comprehensive petrophysical research project to further reservoir

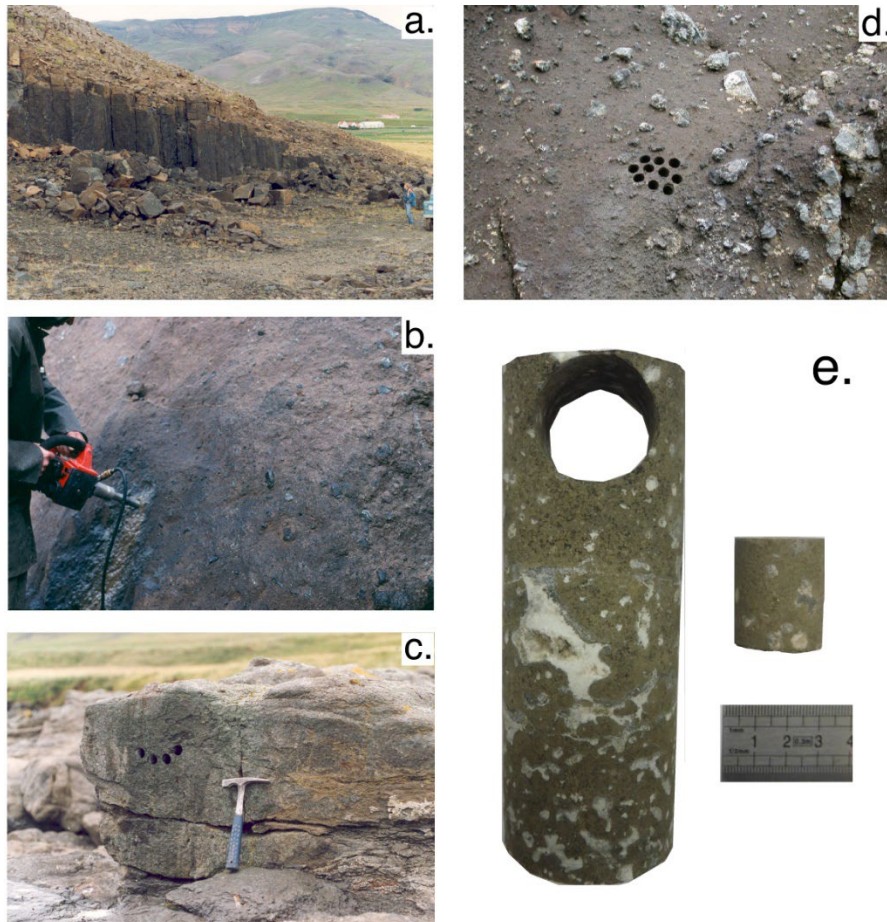

**Figure 1. Photographs of sample collection. a. Basaltic intrusion (dolerite) located on in a quarry on the coast of Hvalfjörður (samples H-90 and H-91), b. Hyaloclastite tuff, located in Námagill (sample G-24), c. Basaltic lava flow, located in Kúludalsá to the south of Akrafjall (sample H-72), d. Hyaloclastite tuff breccia, showing embedded pillow basalt fragments (sample 170803-09). e. Spot core drilled into larger core obtained from 131 m depth in a borehole KH-1 in Krafla (sample L22). The rock is a lava flow altered to smectite-zeolite facies, with vesicles filled mainly with quartz, zeolites and calcite.**

dataset has been useful in elucidating the interrelationship between porosity and permeability (Sigurdsson and Stefansson, 1994; Sigurdsson et al., 2000; Stefansson et al., 1997) and the relationship of these physical properties to the degree of

hydrothermal alteration (Gudmundsson et al., 1995; Franzson et al., 2001; Franzson et al., 2007). This data has also been used to constrain the prior rock property distributions assumed in Bayesian geophysical inversions (Scott et al., 2019) and numerical reservoir models (Scott et al., 2022a). Here, we introduce an updated and expanded version of the database. The goal of this contribution is to ensure that this data remains accessible to future generations of geoscientists and reservoir engineers. In addition to helping constrain numerical models and geophysical inversions, this data can be used to better understand the

interrelationship between lithology, hydrothermal alteration, and petrophysics.

---

modelling by combining petrophysics, geology, alteration and geochemistry. The rock samples used for this research were largely taken at various erosional levels of the Icelandic crust. Just over half of this database is derived from this work.

## 2 Structure and Contents of the Database

*Valgarður* is a publicly accessible database containing petrophysical and chemical/mineralogical analyses of Icelandic rocks. Although many studies have investigated the effect of elevated temperature and pressure on the petrophysical properties of Icelandic rocks (e.g., Vinciguerra et al., 2005; Jaya et al., 2010; Kristínsdóttir et al., 2010; Milsch et al., 2010; Adelinet et al., 2010; Adelinet et al., 2013; Grab et al., 2015; Eggertsson et al., 2020a,b; Nono et al., 2020; Kummerow et al., 2020; Weaver et al., 2020, Heap and Violay, 2021), at present we restrict the database to measurements at near-ambient conditions, in order to facilitate comparison between the different studies and ensure consistency among the reported data.

Sample collection involves drilling a ~2.5 cm diameter plug of variable length into a surface outcrop or section of core (Figure 1). Variability in the sample collection process and analytical methods is to be expected given the long period over which the underlying data comprising the database were collected. During assembly of the database, we sought to ensure that the results of the different studies are reported in a consistent manner. As different studies used different methods to analyze given petrophysical properties, each data point is accompanied by a description of the methodology or origin of the data.

Table 1 shows a description of the sources of the data for the database. The original 529 samples collected by Orkustofnun and Iceland Geosurvey (ÍSOR) between 1990-2010 that made up the first release of the Valgardur database mainly originate from hand-drilled cores taken at the surface within the neovolcanic zone or at erosional surfaces in the older strata at various palaeo-depths and alteration stages (Sigurðsson and Stefánsson, 1994; Guðmundsson et al., 1995; Franzson et al., 2007; Friðleifsson & Vilmundardóttir, 1998; Franzson et al., 2011). In this release of the database, we added data from 302 samples collected by Orkustofnun between 1970-1980 (Pálsson, 1972; Pálsson et al., 1984), 161 samples from downhole cores obtained from active geothermal systems (Flovenz et al., 2005; Franzson & Tulinius, 1999; Reinsch et al., 2016; Bär et al., 2020; Lévy et al., 2018, 2019a, 2019b, 2020a, 2020b; Gilbert et al., 2020; Nono et al., 2020), 92 samples obtained from boreholes drilled during evaluation of a hydropower project in the Búðarháls area (Árngrímsson and Gunnarsson, 2009), as well as 31 new analyses from borehole samples from the Theistareykir geothermal area and 3 surface samples from the Austurhorn gabbro.

To facilitate simple and user-friendly handling, the database is provided in 'flat' format (one row per sample) rather than 'stacked' format (one row per measurement). The main Excel file is divided into two worksheets:

1. Petrophysical properties
2. Mineralogical and geochemical properties

The first and primary table reports measurements of petrophysical properties, including porosity, grain density, permeability, electrical resistivity, acoustic velocities, rock strength, and thermal conductivity performed at near-ambient conditions. This table provides lithologic characterization, including detailed sample descriptions in both Icelandic and English, and description of alteration zone. This table also reports detailed sample metadata including sample type (surface or borehole), date and location of sample collection. Following the example of Bär et al. (2020), we provide information about how each measurement was acquired in a 'Remarks' column adjacent to the reported value and set the fill colour of cells based on the type of data

contained in the cell (e.g., cells listing the primary and secondary references are coloured yellow, cells containing sample meta-data are coloured blue, cells related to lithological characterization are coloured orange, etc.).

The second table reports geochemical and mineralogical data. The data reported on this worksheet includes petrographic observations (point-counting on thin sections), bulk rock geochemical analyses derived from X-ray fluorescence (XRF) analyses, or quantitative mineralogical assessments using X-ray diffraction (XRD). Although many studies have investigated the geochemistry and petrology of Icelandic rocks (e.g. Sigmarsson and Steinthórsson, 2007; Sigmarsson et al., 2008) and much of the available data has been compiled into a publicly-accessible database (Harðardóttir et al., 2022), we restrict the

scope of the database and provide geochemical and mineralogical data for samples that also have petrophysical properties given in the first table.

**Table 1. Description of sources of data comprising the Valgarður database.**

| References | Description | Number of samples |
|---|---|---|
| Pálsson (1972); Pálsson et al., (1984); Friðleifsson (1973, 1975, 1978) | Early studies of petrophysical properties (grain density, connected/total porosity). Mainly comprises samples obtained at surface or from shallow boreholes. Scant description of geology and alteration; samples are described as 'Unaltered' or 'Altered', rather than an alteration zone as listed in Table 3. | 339 |
| Sigurðsson and Stefánsson (1994); Guðmundsson et al. (1995); Sigurðsson (1998a,b); Sigurðsson et al. (2000); Sigurðsson and Stefánsson (2002); Franzson et al. (1997); Franzson et al. (2001); Franzson et al. (2008) | Systematic studies of rock properties in fossil and active geothermal systems. Includes petrophysical (grain density, total/connected porosity, permeability) as well as geochemical and petrographic data. | 351 |
| Friðleifsson & Vilmundardóttir (1998); Guðlaugsson (2000) | Detailed study of a single lava flow in the Reykjavik area. Samples taken at different depth levels within lava flow to see variation in petrophysical properties. Includes grain density, total/connected porosity, permeability, whole rock geochemistry, acoustic velocities, thermal properties as well as petrographic observations. | 85 |
| Franzson & Tulinius (1999) | Borehole samples from ÖJ-1 (in Ölkelduháls in the Hengill area). Samples obtained from altered hyaloclastite tuff at ~800 m depth. Includes electrical properties as well as grain density, total/connected porosity, permeability, whole rock geochemistry, and point counting. | 14 |
| Frolova et al. (2005); Franzson et al. (2010); Frolova (2010); Franzson et al. (2011) | Hyaloclastite tuff mainly obtained from surface outcrops in southwest Iceland. Most samples show a low degree of alteration. Includes grain density, total/connected porosity, permeability, | 101 |

| | | |
|---|---|---|
| | whole rock geochemistry, acoustic velocities, and mechanical properties. | |
| Flóvenz et al. (2005) | Investigation of the effect of alteration on the electrical properties of geothermal reservoir rocks. Borehole samples obtained from Krafla, Hengill, and Reykjanes. Includes electrical acoustic properties as well as grain density and connected porosity. | 12 |
| Arngrímsson and Gunnarsson (2009); Foged and Andreassen (2016) | Borehole samples retrieved during evaluation of tunnelling and hydropower activities in the Búðarháls area. Several samples are relatively soft silicic, altered, and sedimentary rock formations. Includes grain density, connected porosity as well as mechanical data from unconfined compression, triaxial compression and Brazilian disk tests. | 92 |
| Reinsch et al. (2016); Nono et al. (2020); Bär et al. (2020) | Study of rock properties in fossil and active geothermal systems conducted as part of the IMAGE project. Includes grain density, connected porosity and permeability as well as electrical and acoustic properties. | 20 |
| Lévy et al. (2018, 2019a, 2019b, 2020a, 2020b) | Krafla core samples from research wells KH-1, KH-3, KH-5, and KH-6. Includes connected porosity and grain density (triple weight), permeability, electrical properties, and acoustic velocities. Quantitative mineral characterization using XRD. | 94 |
| Gilbert et al. (2020) | Samples obtained from IDDP-2 at ~3.6-4.6 km depth. Includes grain density and connected porosity (triple weight) as well as electrical conductivity and acoustic properties. | 20 |
| Present study | Borehole samples from well ÞR-07 in Theistareykir. Connected porosity, grain density (triple weight), electrical properties, and acoustic velocities. Quantitative mineral characterization using XRD. | 31 |
| Present study | Surface samples from the Austurhorn gabbro in SE Iceland. Connected porosity, grain density (triple weight), electrical properties, and acoustic velocities. | 3 |

In addition, we provide two additional files in the repository:

1. A zip file containing photographs of many of sampling sites. A file listing the names of files which correspond to the sample IDs is also provided in the repository. These photographs have been included with the database to facilitate future investigations in the study areas.

2. An extended references worksheet listing all the references referred to as primary or secondary references in the
database.

## 2.1 Sample ID and References

The order in which samples are presented in the database is approximately chronological, based on the date of the reference and the order of the sample numbers within the reference. The sample ID is equivalent to the reported sample ID in the primary reference. Primary references indicate where the data was first published and/or best-documented; secondary references are also given if the data was reported or used in further studies, or in the case of borehole samples, describe core logs reported after the drilling of the well. References that are not cited in this text but are cited in one of these columns are described in the references worksheet provided in the Excel database.

## 2.2 Location Coordinates and Description

A description of the sampling location is available for all samples. The location description is provided in Icelandic or English and is reported according to a geographic feature (mountain, lake, stream, etc.) or village/town. In the case of core samples, the name of the geothermal field and well from which the sample is obtained is given. Precise location coordinates describing the location of the samples are generally only available for samples collected after ca. 1995. For samples without location coordinates given in the primary reference (Pálsson, 1972; Pálsson et al., 1984), an approximate sampling location was estimated based on the sample location description. Therefore, there is significant uncertainty (up to 0.5 km or more) in the location of samples collected before 1995. For many of the samples, photographs were taken which show in some detail the location and the geological features (Figure 1). These photographs are provided in a supplement to the database.

The latitude and longitude of the sampling point at the surface in decimal degrees is reported according to the reference system WGS84 as well as in projected coordinates in ISNET93, the latter being widely used in Iceland. The elevation is given in meters above sea level (m a.s.l.) and was obtained using Google Elevation services when not provided in the primary reference. Borehole samples show the coordinates at the wellhead and an accompanying depth (in meters). The uncertainty in elevation and depth coordinates is estimated to be on the order of 1-10 m for most samples, the uncertainty is significant greater (0.1-0.5 km) for the samples collected prior to 1990 (Pálsson, 1972; Pálsson et al., 1984).

Figure 2 shows the locations of all obtained surface and borehole samples. There is a greater abundance of samples from the southwest of Iceland, including the area around Reykjavik, Akranes, Borganes, and the Reykjanes peninsula. Many of the samples are taken at deep erosional levels within fossil geothermal systems, including the Geitafell central volcano located in the Hornafjörður region in the southeast (Friðleifsson, 1983a,b; Friðleifsson, 1984), the Hafnarfjall-Skarðsheiði central volcano located in the west (Franzson, 1978), and the Esja volcanic region located close to Reykjavik (Friðleifsson, 1973). There are several surface samples of altered volcanic rocks collected from active geothermal areas, including the Reykjanes peninsula (Svartsengi, Krýsuvík, and Reykjanes), the Hengill region, and Landmannalaugar. Borehole samples are available from the major active geothermal areas (Hengill, Reykjanes, Krafla, and Theistareykir), as well as from several wells drilled outside of thermal areas during the evaluation of hydropower projects in Fljótsdalshreppur and Hrauneyjar (Pálsson et al., 1972, 1984; Arngrímsson and Gunnarsson, 2009).

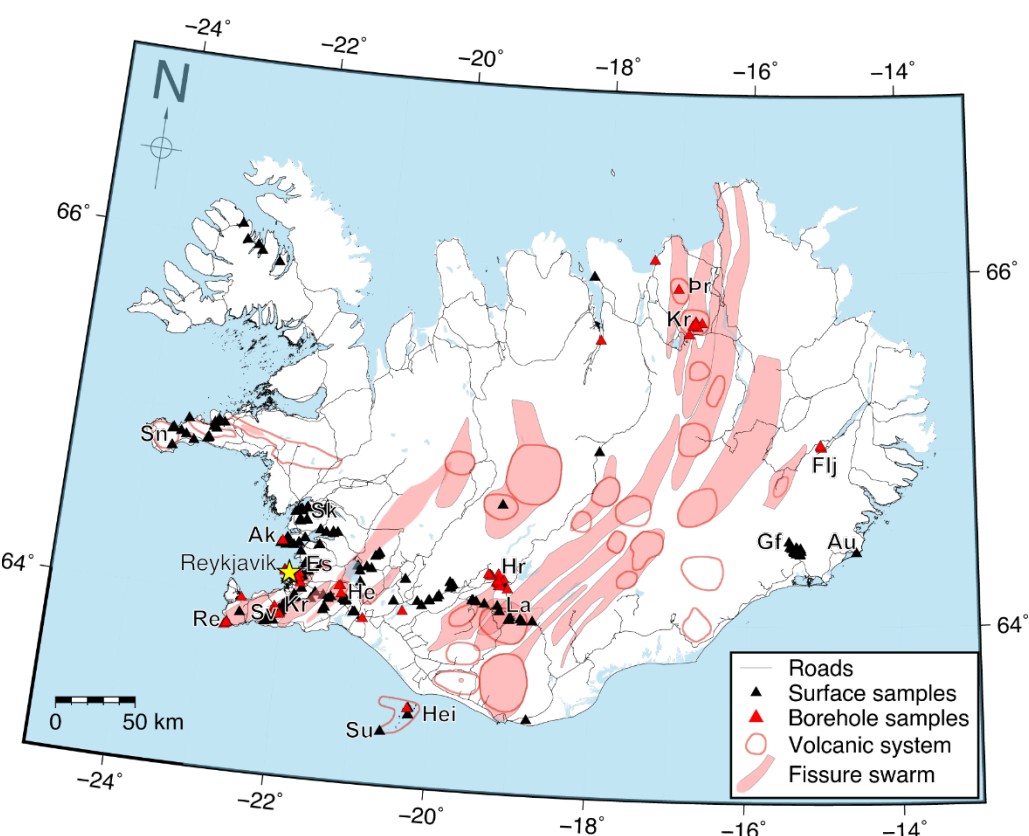

**Figure 2. Map of Iceland showing the locations of surface (black triangles) and borehole samples (red triangles) comprising the database. Volcanic systems outlined with thick red lines and associated fissure swarms highlighted in light red. Roads are shown as thin black lines. Locations mentioned in text: Ak = Akranes, Au = Austurhorn, Es = Eja, Flj = Fljótsdalshreppur, Gf = Geitafell, He = Hengill, Hei = Heimaey, Hr = Hrauneyjar, Hv = Hvalfjörður, Kr = Krafla, Kv = Krýsuvík, La = Landmannalaugar, Re = Reykjanes, Sk = Skarðsheiði, Sn = Snæfellsness, Su = Surtsey, Sv = Svartsengi, Þr = Theistareykir. Incorporates data from the Icelandic Institute of Natural History (IINH, 2022).**

## 2.3 Rock type characterization

The sample description provides a summary of characteristics on the scale of the hand sample, including but not limited to
grain size, color, vesicle size, the presence of layering, fractures, joints and fissures, and relevance to other samples. Sample
descriptions that were originally provided only in Icelandic were translated into English and are listed in separate columns.
The level of detail of sample description varies between the different studies. Each sample is assigned to one of eight broad
lithological categories and 24 more detailed lithological identifiers, following the classification scheme of Guðmundsson et al.
(1995) (Table 2). Lithological identifiers were determined based on the interpretation of the geological context and visual
characteristics, rather than whole-rock chemical analyses. For borehole samples, the sample description is obtained from the
description of the core log at the logged depth.

**Table 2. List of lithological identifiers, detailed description of rock types covered by that identifier, and number of samples in the database corresponding to each lithology.**

| Broad lithological category | Lithological identifier | Number of samples |
|---|---|---|
| Lava flow | Flow-top breccia | 34 |
| | Fine-medium grained basaltic lava | 247 |
| | Medium-coarse grained basaltic lava | 105 |
| | Porphyritic basaltic lava | 109 |
| | Total | 495 |
| Hyaloclastite | Hyaloclastite breccia | 77 |
| | Hyaloclastite tuff | 178 |
| | Hyaloclastite sediment | 12 |
| | Total | 267 |
| Pillow basalt | Pillow basalt | 42 |
| Silicic volcanic | Rhyolite lava | 22 |
| | Rhyolite hyaloclastite | 35 |
| | Rhyolite tuff | 10 |
| | Rhyolite breccia | 22 |
| | Ignimbrite | 7 |
| | Total | 96 |
| Intermediate volcanic | Basaltic andesite | 6 |
| | Icelandite (andesite) | 13 |
| | Dacite | 2 |
| | Total | 21 |

| | Fine-medium grained basaltic intrusion | 54 |
|---|---|---|
| Basaltic intrusion | Medium-coarse grained basaltic intrusion | 53 |
| | Porphyritic basaltic intrusion | 16 |
| | Gabbro | 16 |
| | Total | 139 |
| Silicic intrusion | Rhyolite dyke | 17 |
| | Granophyre | 7 |
| | Total | 24 |
| Intermediate intrusion | Diorite | 6 |
| Sediment | Sediment | 66 |

Among extrusive basaltic volcanic rocks, one key distinction is between lava flows, which are erupted sub-aerially, and hyaloclastites, which are often erupted sub-glacially (e.g., Schopka et al., 2006), but may crystallize in a mixture of sub-aqueous and partially sub-aerial conditions (Bergh and Sigvaldason, 1991; Banik et al., 2014). While fresh olivine basaltic lava flows are referred to as *grágrýti* in Icelandic, tholeiitic lava flows, evolved from more mafic olivine basalt, are often

referred to as *blágrýti*. The latter usually show flow-banding and are finer grained than the olivine basalts. It has been customary in Iceland to classify lava flows based on crystal size (e.g., Guðmundsson et al., 1995); in comparison to the classification scheme of Walker (1958), the lithological identifier 'fine-medium grained basaltic lava' generally corresponds to the 'tholeiitic basalt' type, while 'medium-coarse grained basaltic lava' generally corresponds to the 'olivine basalt' type. Porphyritic basalts show macroscopic (up to 2 cm in diameter) plagioclase phenocrysts, with less abundant pyroxene and

olivine phenocrysts (Fig. 1c). Basaltic lava flows are often vesicular, especially towards the tops of the individual lava flow units, where they develop a thick surface rubble or a ropy texture. Such units are referred to as flow-top breccias (*kargi*) or entablature/cube-jointed basalt (*kubbaberg*), which often shows irregular columnar jointing.

Hyaloclastite formations often contain denser pillows or pillow fragments embedded in a tuff matrix (Figure 1d). Hyaloclastite (*móberg*) contains a higher proportion of glassy material compared to lava flows. Hyaloclastite breccias and pillow basalts

show significant heterogeneity on the scale of cm-m. To maximize the homogeneity among the cores drilled for a given rock sample, hyaloclastite samples were obtained from the dominantly glass-rich tuff matrix (e.g Fig. 1d). Due to the strong contrast in the physical properties of the pillow basalt fragments compared to the tuffaceous matrix, pillow basalts comprise a separate lithological category.

Silicic volcanic rocks are often found as 50-100 m thick flows in the vicinity of volcanic vents. Silicic volcanic products

include vesicular glassy pyroclastics such as pumice, obsidian, which forms from rapid cooling at the margins of rhyolitic lava flows, perlite, a glassy variety of rhyolite with high water content, and ignimbrite, which forms during explosive eruptions

when volcanic material cascades down slope as pyroclastic density currents. Intermediate volcanic rocks include basaltic andesite, icelandite (which is considered interchangeable for andesite) as well as dacite.

Among intrusive rock types, basaltic intrusions are distinguished from silicic or intermediate intrusions. Basaltic intrusions include both gabbro, which crystallizes in larger subsurface magma bodies, as well as dolerite (sometimes referred to as diabase), which is found in basaltic dykes, including cone-sheets. Silicic intrusions include rhyolite dykes, granophyre, the latter usually showing fine-scale intergrowth of quartz and feldspar, as well as microgranite, which lacks granophyric intergrowths. Intermediate intrusive rocks including diorite are relatively rare. Note that intrusions may be of very variable age in relation to the geothermal system they intrude.

While many volcaniclastic rocks (e.g., hyaloclastite) show sedimentary textures related to deposition in a sub-aqueous or sub-aerial environment (Bergh and Sigvaldason, 1991; Schopka et al., 2006; Banik et al., 2014; Greenfield et al., 2020), they are not referred to as sedimentary rocks in the database. Examples of sedimentary rocks found in Iceland include clay-rich lacustrine sediments, glacial tillite and conglomerates, sandstone, as well as interbasaltic beds (e.g., Bennet et al., 2000; Arnalds et al., 2001; Thorpe et al., 2019; Eiriksson and Simonarson, 2021). However, distinction between these rock types is not made in the database (Table 2) due to the emphasis on volcanic and igneous rocks. It should though be noted that sedimentary grains are near exclusively of igneous origin.

In addition to the lithological classification, each sample is assigned to one of the main alteration zones identified in Icelandic rocks (Table 3). Most of alteration minerals in Iceland fall in the $Ca^{2+}$ (stilbite-heulandite-laumontite-wairakite) and $Ca^{2+}+Mg^{2+}+Fe^{2+}$ (smectite-chlorite-epidote-actinolite) series of minerals (Walker, 1960; Walker, 1974; Kristmannsdóttir and Tómasson, 1978; Kristmannsdóttir, 1979; Lonker et al., 1993; Franzson and Gunnlaugsson, 2020; Escobedo et al., 2021). With increasing depth and temperature, the alteration zones are the smectite-zeolite zone, the mixed-layer clay zone, the chlorite-epidote zone, the epidote-actinolite zone, and the amphibole zone (Kristmannsdóttir 1979; Sveinbjörnsdóttir 1992). In some studies (e.g., Franzson et al., 2008), further distinction is made between a chlorite zone and the chlorite-epidote zone; for this study, we combine these two zones for simplicity and to facilitate comparison among the different studies. In addition, we combine the amphibole zone with the epidote-actinolite zone. Although rocks without obvious alteration mineralogy are described as unaltered, the basaltic glass has undergone some extent of palagonitization. Palagonitization occurs as post-eruptive process entailing the hydration of basaltic glass and replacement by secondary minerals, including zeolites and smectites (Stroncik and Schminke, 2002). Although this is a type of alteration process, it does not correspond to a specific alteration zone as observed in active and fossil geothermal systems. Variable porosity and permeability have a pronounced effect on the alteration intensity of the rock.

**Table 3. Description of alteration zones and number of samples in the database corresponding to each alteration zone.**

| Alteration zone | Description | Number of samples |
|---|---|---|
| Unaltered | Rocks without obvious alteration mineralogy. Most glass-rich hyaloclastite rocks have undergone palagonitization but are still classified as unaltered. | 510 |
| Smectite-zeolite | Replacement of basaltic glass and olivine by smectite clay (mostly saponite). Zeolite minerals precipitate in open vesicles but are also found dispersed in replaced glass. Occurs at temperatures below 200 °C. Often coexists with chalcedony. | 262 |
| Mixed-layer clay | Interlaying of smectite and chlorite occurs to an increasing extent at 200-230 °C. Onset of plagioclase alteration. In a more acidic environment, mixed-layer smectite-illite is observed but is rare. | 88 |
| Chlorite-epidote | Chlorite is the dominant sheet silicate at rock temperatures >230 °C. Epidote occurs sporadically >240 °C but may precipitate in larger quantities at high permeability. Often coexists with prehnite and wairakite. | 159 |
| Epidote-actinolite | High-grade greenschist facies assemblage. Actinolite forms in fine-grained aggregates together with chlorite and epidote at temperatures >280 °C. May include secondary pyroxenes or feldspars (albite) at higher temperatures. The zone includes also wollastonite. | 113 |

## 3 Data Sources

Measurements at ambient conditions of room temperature and atmospheric pressure include grain density, porosity, electrical conductivity, and acoustic velocities. Permeability measurements were made under a variable but low confining pressure (<5 MPa). Table 4 shows the number of samples with data corresponding to the different petrophysical and mineralogical properties. Only a few samples were analyzed for mechanical properties. Depending on the source of the data, different analytical techniques were used to measure a given petrophysical property. This can make it challenging to report the measured quantities in a consistent manner.

**Table 4. Number of samples with both petrophysical/mineralogical data for different properties in the database. Bold text indicates the number of samples analysed for a single property. XRF = X-ray fluorescence for bulk whole-rock geochemistry, XRD = X-ray diffraction for quantitative mineralogical composition.**

| | Connected porosity | Total porosity | Point-counting | Grain density | Gas perm. | Liquid perm. | Intrin. perm. | Elec. resist. | XRF | XRD | Acoustic veloc. | Strength | Thermal conduct. |
|---|---|---|---|---|---|---|---|---|---|---|---|---|---|
| Connected porosity | **1160** | 512 | 356 | 1143 | 501 | 102 | 498 | 176 | 349 | 124 | 293 | 91 | 54 |
| Total porosity | 512 | **514** | 158 | 514 | 130 | 36 | 128 | 9 | 191 | 0 | 0 | 0 | 2 |
| Point-counting | 356 | 158 | **361** | 355 | 289 | 41 | 287 | 10 | 288 | 0 | 135 | 54 | 53 |
| Grain density | 1143 | 514 | 355 | **1144** | 496 | 97 | 492 | 177 | 342 | 120 | 287 | 91 | 54 |
| Gas permeability | 501 | 130 | 289 | 499 | **501** | 87 | 497 | 55 | 263 | 40 | 186 | 53 | 54 |
| Liquid permeability | 102 | 36 | 41 | 97 | 87 | **102** | 87 | 51 | 45 | 45 | 47 | 0 | 2 |
| Intrinsic permeability | 498 | 128 | 287 | 492 | 497 | 87 | **498** | 55 | 260 | 41 | 186 | 53 | 53 |
| Electrical resistivity | 176 | 9 | 10 | 177 | 55 | 51 | 55 | **177** | 10 | 119 | 146 | 0 | 2 |
| XRF | 349 | 191 | 288 | 342 | 263 | 45 | 260 | 10 | **354** | 0 | 75 | 47 | 15 |
| XRD | 124 | 0 | 0 | 120 | 40 | 45 | 41 | 119 | 0 | **124** | 124 | 0 | 0 |
| Acoustic velocities | 293 | 0 | 135 | 287 | 186 | 47 | 186 | 146 | 75 | 124 | **295** | 54 | 52 |
| Strength | 91 | 0 | 54 | 91 | 53 | 0 | 53 | 0 | 47 | 0 | 54 | **92** | 2 |
| Thermal conductivity | 54 | 2 | 53 | 54 | 54 | 2 | 53 | 2 | 15 | 0 | 52 | 2 | **54** |

## 3.1 Porosity and Grain Density

Porosity is differentiated between connected porosity (the fraction of bulk volume occupied by pore space connected to the outside surface of the sample; this is also referred to as "effective porosity") and total porosity (the fraction of bulk volume occupied by pore space). In igneous and volcanic rocks, gas bubbles may form unconnected pores, particularly when volatile content is low, and porosity may be largely unconnected when porosity is less than ~0.1 (Colombier et al., 2017). Different analytical methods are available for quantifying porosity; among the most widely used methods are gas expansion (He pycnometry) and saturation/imbibition methods (Anovitz and Cole, 2015). While measurement of connected porosity using methods such as triple-weighting is non-destructive, determination of total porosity requires crushing the sample to measure the density of the solid material via conventional methods such as Hg displacement.

Gas expansion methods are based on Boyle's law and the ideal gas law. A gas, usually He due to its ability to penetrate narrow pore throats (>1 nm; Anovitz and Cole, 2015), expands isothermally from a reference cell at a known pressure into the sample

container. The resulting equilibrium pressure reflects the volume of the pores into which the He gas has penetrated, calculated using Boyle's law. As the bulk volume of the sample $V_{bulk}$ is known based on the geometry of the sample, connected porosity can be calculated following Eq. (1):

$$\phi_{connected} = \frac{V_{pore}}{V_{bulk}} \tag{1}$$

where $V_{pore}$ is the fraction of interconnected pore space.

Saturation/imbibition methods are based on weighing a dry sample prior to full saturation with a wetting fluid ($W_{dry}$), after immersing the fluid in a saturating fluid for an extended period ($W_{immersed}$), and again after removing excess liquid from the surface of the sample ($W_{sat}$). The porosity is then given by Eq. (2):

$$\phi_{connected} = \frac{V_{bulk} - V_{matrix}}{V_{bulk}} = \frac{V_{bulk} - (W_{dry} - (W_{immersed} - W_{crad}) + W_{crad})/\rho_{fluid}}{V_{bulk}} \tag{2}$$

where $\rho_{fluid}$ is the density of the saturating fluid and $W_{crad}$ is the weight of the cradle used to immerse the sample.

Connected porosities measured using gas expansion and triple weighing methods yield similar results, at least within a margin of uncertainty of <2-5%. This is demonstrated in Figure 3a, which reports connected porosity data collected by both methods on core samples from Krafla (Lévy et al., 2018, 2020b). For gas expansion measurements, an additional source of uncertainty

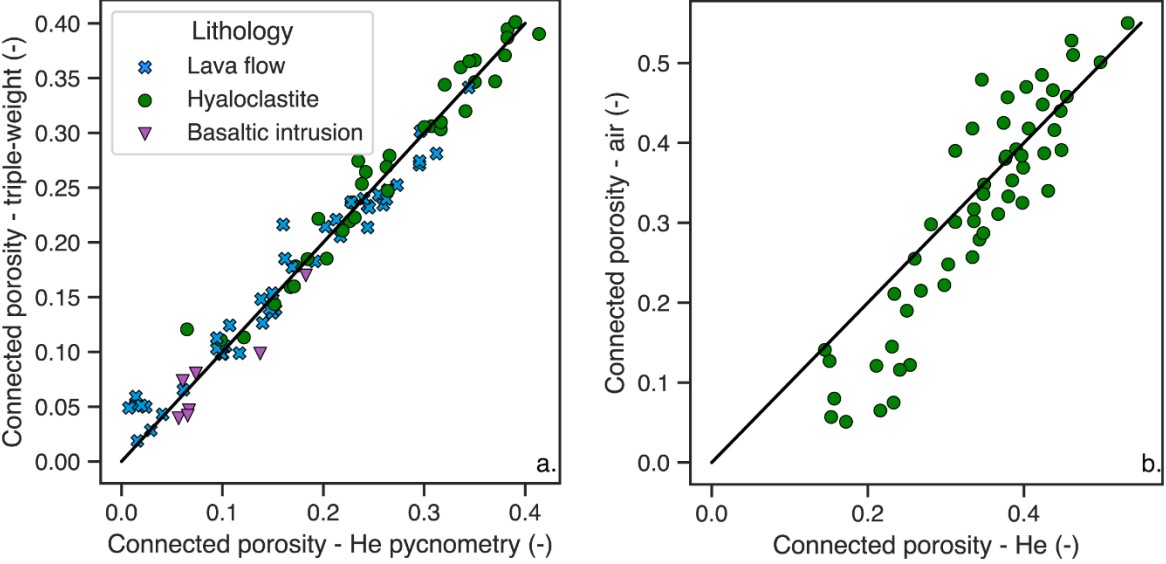

**Figure 3. Comparison of connected porosity measurements using different measurement techniques. A. Connected porosity obtained by helium expansion and versus connected porosity measured using triple weighting. Data derived from a subset of core samples originating from Krafla (Lévy et al., 2018, 2020b). b. Comparison of connected porosity measurements on hyaloclastite tuffs using He or air as the saturating gas. Data is from Franzson et al. (2011).**

is related to the choice of the saturating gas. Figure 3b compares connected porosity measurements performed on a set of hyaloclastite samples using either He (Franzson et al., 2011) or air (Frolova et al., 2005) as the saturating gas, and shows that connected porosity of samples measured using air is lower at low porosities and higher at high porosities. While the former may be due to the lesser ability of air to penetrate the microporosity, the latter may result from adsorption of water contained in the air in the clay-rich, altered rock. An additional source of error is that helium (or air) pycnometry requires the sample dimensions, whereas the triple-weight method only uses measurements of weight. As laboratory measurements of weight are often more accurate than measurements of length, this is one advantage of the triple-weight method over helium pycnometry.

However, repeat measurements of connected porosity on different core plugs obtained from a given rock outcrop (e.g. Fig. 1d) reveal natural uncertainty in the sampled rock can exceed 5-10% (i.e., connected porosity can range from 5-15%). Therefore, for volcanic rocks such as hyaloclastites or lava flows, which can show strong gradients in petrophysical properties over distances <1 m, the uncertainty resulting from different measurement devices and methods is likely less than (or comparable to) the natural variability present in the rock.

All of the total porosity and most of the grain density data reported in this database were determined by pulverizing the sample and measuring the density of crushed materials using conventional techniques (e.g. Hg displacement). Total porosity can be calculated following Eq. (3):

$$\phi_{total} = \frac{V_{bulk} - V_{grain}}{V_{bulk}} = \frac{\rho_{grain} - \rho_{bulk}}{\rho_{grain}} \tag{3}$$

As noted in Colombier et al. (2017), although accurate measurement of the density of the solid, pore-free phase(s) in the volcanic rock is required to calculate total porosity, heterogeneity in the phenocryst assemblage between clasts or variations in bulk composition may be common. Grain density can also be approximated based on connected porosity measurements from triple-weighting following Eq. (4):

$$\rho_{grain,TW} = \frac{W_{dry}}{V_{matrix}} = \frac{W_{dry}}{V_{bulk} - \frac{(W_{sat} - W_{dry})}{\rho_{fluid}}} \tag{4}$$

However, if there is a significant fraction of unconnected porosity, Eq. (4) will systematically overestimate grain density. Figure 4a shows that the distribution of grain density of smectite-zeolite altered lava flows is similar whether measured by Hg displacement (blue lines) or triple-weighting (red lines) techniques. On the other hand, for altered hyaloclastites (Fig. 4b), samples analyzed by triple-weighting ($\sim$2.75 g cm$^{-3}$) show significantly larger average grain density than by Hg displacement ($\sim$2.6 g cm$^{-3}$). However, other factors might also explain this discrepancy, most notably that many of the samples shown in Fig. 4b analyzed using Hg displacement were derived from surface outcrops, whereas those analyzed by triple-weighting were obtained from borehole samples in active geothermal systems. Although these samples are in similar alteration zones, rocks at depth are more compacted, and may contain a denser alteration mineral assemblage.

Porosity was also assessed in 352 samples by point-counting. In point-counting, a thin section (around 30 micrometers thick) of the sample was prepared, and a regular grid with a given number of points (usually 200 or 1000 points) was arrayed onto the thin section image. Identification of mineralogy or pore space at each point was performed, with the different studies

applying different levels of classification between primary minerals, glass, pore space, and alteration minerals. The primary porosity formed during magma emplacement and cooling was estimated as the sum of remaining open-space porosity in the rock as well as that of secondary alteration minerals that have precipitated into vesicles (Petford, 2003). This technique does not measure the cross-sectional area of microcracks, but rather only identifies macroscopic pores on the order of 1 mm or larger (Neuhoff et al., 1999; Manning and Bird, 1995; Chayes, 1956). Although the contribution of the latter to the porosity is

often large, note that distinction between open-space porosity created by post-eruptive processes (fracturing/veining) was not made by all studies. Therefore, we estimate that the uncertainty of porosity measurements by point counting is considerable (>5-10%). Figure 5 shows that the remaining open-space porosity measured by point counting is generally lower than that obtained by gas expansion, particularly for altered, high-porosity hyaloclastites, indicating the dominance of microporosity on the total porosity. However, a significant number of samples (~50) had higher porosity recorded using petrographic analysis

than by gas expansion. As porosity and permeability are scale-dependent (Manning and Bird, 1995), such variability could

also indicate natural heterogeneity in the rock and preparation of the thin section from a particularly compact or porous part of the material.

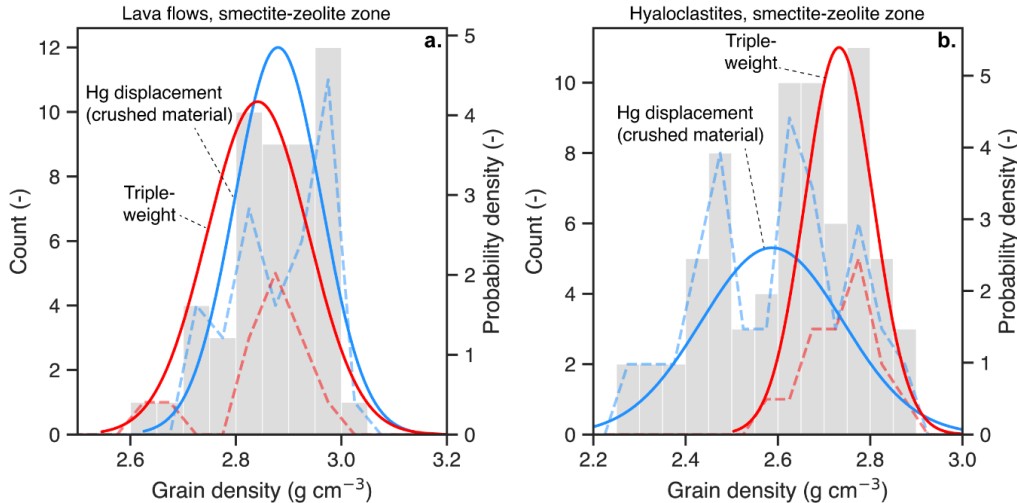

**Figure 4. Grain density obtained by mercury displacement (blue dashed lines) and triple-weighting (red dashed lines) for (a) lava flows and (b) hyaloclastites. All data is shown in grey and a normal distribution fit to the data for each respective measurement technique is shown with solid blue or solid red lines. Note that all of the samples for which grain density was measured by triple-weighting were core samples, while most of the samples for which grain density was measured using Hg displacement were surface samples. This could affect the difference seen in (b).**

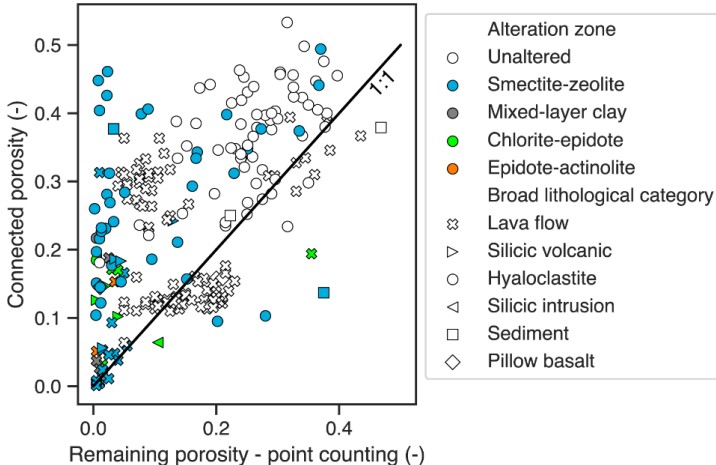

**Figure 5. Remaining porosity estimated by point counting compared with connected porosity measured using He pycnometry.**

## 3.2 Permeability

Permeability is measured using a gas, usually helium or argon, or, more rarely, using water (Heap et al., 2018). Many of the samples in Guðmundsson et al. (1995) were measured at Core Laboratories (formerly Western Atlas Core Laboratories) using the CMS-300 device, consisting of a gas cylinder with a known volume, a pressure sensor, and a core holder that can be opened into a gas cylinder charged with helium gas, generally up to 1.5 MPa, and the atmosphere. While permeability of the samples from Guðmundsson et al. (1995) were measured using a low (<4 MPa) confining pressure, measurements performed on a few samples under varying confining pressure showed little dependence of permeability on confining pressure (Johnson and Boitnott, 1998). Samples from Lévy et al. (2020) were measured at University of Montpellier using a steady-state method on cylindrical samples confined at a pressure of 4 MPa. A constant argon pressure was imposed at the sample inlet and pressure at the sample outlet was maintained at 1 bar, and argon flow rate were measured by a gas flow-meter. For each sample, the argon pressure at the inlet was systematically varied from about 5 to 40 bars to evaluate the intrinsic permeability as described below. In the water permeability measurements, a servo-controlled pump maintained constant pore pressure constant and the flow rate was determined using the measured piston displacement. For samples with permeability lower than $10^{-19}$ m$^2$, a pulse decay method was used (e.g., Brace et al., 1968).

As pressure at the sample inlet and outlet of the core is known and the flow through the core is proportional to the pressure drop, Darcy's law is then used to calculate permeability from the above quantities:

$$k = -\frac{q\mu}{A\left(\frac{dp}{dx}\right)} \tag{5}$$

where $k$ is the rock permeability, $q$ is the volumetric flow rate of the gas or liquid, $\mu$ is the viscosity of the fluid, $A$ is the cross-sectional area, and $\frac{dp}{dx}$ quantifies the pressure drop along the core.

When using gas or liquid to measure permeability in rock samples, it is necessary to correct the measured permeability for systematic measurement error resulting from the sample geometry and the properties of the fluid (Heap et al., 2018). In low permeability rocks, interactions between the fluid molecules/atoms and the pore walls reduce resistance to flow. Intrinsic permeability is calculated from measured gas permeability at a range of gas pressures using the Klinkenberg correction (Klinkenberg, 1941), which is based on the following equation:

$$k_i = \frac{k_g}{\left(1+\frac{b}{p_m}\right)} \tag{6}$$

where $k_g$ is the measured gas permeability calculated from equation (5), $p_m$ is the average gas pressure at which $k_g$ is measured, and $b$ is the Klinkenberg coefficient, taken as a constant for a certain gas and a certain rock. The Klinkenberg coefficient and intrinsic permeability $k_i$ are obtained by plotting measured gas permeability at a range of pressures against $\frac{1}{p_m}$, with the slope corresponding to $k_i$ and the y-intercept $b$ (when $p_m$ goes to infinity). The Klinkenberg coefficient depends on various properties of the rock, particularly the geometry of the pore space, and is generally most significant in low-porosity,

low-permeability rocks (Filomena et al., 2014). As a result, relative measurement accuracy is greater (±5%) for high permeability rocks ($\geq 10^{-14}$ m$^2$) and reaches up to ±400% in low permeability rocks ($\leq 10^{-16}$ m$^2$). However, in high permeability rocks, turbulent flow regimes may develop, and Darcy's law needs to be modified to take into additional flow resistance resulting from inertial forces, as given by the second term in the Forchheimer equation (Forchheimer, 1901; Zeng and Grigg, 2006):

$$\frac{dp}{dx} = \frac{\mu}{k}\frac{q}{A} + \beta\left(\frac{q}{A}\right)^2 \qquad (7)$$

For the samples analyzed by Guðmundsson et al. (1995), the inertial coefficient $\beta$ was calculated for high permeability (>10$^{-14}$ m$^2$) samples using repeat measurements at different flow rates. For the samples analyzed by Levy et al. (2018), the Forchheimer correction was not applied, and therefore the reported permeability may not be the actual rock permeability; however, as many of these samples are low permeability (<10$^{-15}$ m$^2$), turbulent conditions are unlikely to develop and the

Forchheimer correction is likely not significant (Heap et al., 2018).

Measured permeabilities listed in the database range over nine orders of magnitude, from ~$10^{-20}$ m$^2$ to ~$2\times10^{-11}$ m$^2$. Figure 6a shows that apparent permeability measured using air generally exceeds that measured using water, often by several orders of magnitude.  While this difference is often attributed to water adsorption on clays (see, for example, Tanikawa and Shimamoto, 2009), even in clay-free volcanic rocks, liquid permeabilities can also be lower than gas permeabilities due to water adsorption

on narrow, tortuous microstructural elements (Heap et al., 2018). Measurements of air permeability that are less than brine permeability are generally considered unreliable. Figure 6b compares air apparent permeability and intrinsic permeability, showing that the magnitude of the Klinkenberg correction increases with decreasing permeability, consistent with increased gas slippage during flow through microstructural elements in low-porosity, low-permeability rock (Heap et al., 2018).

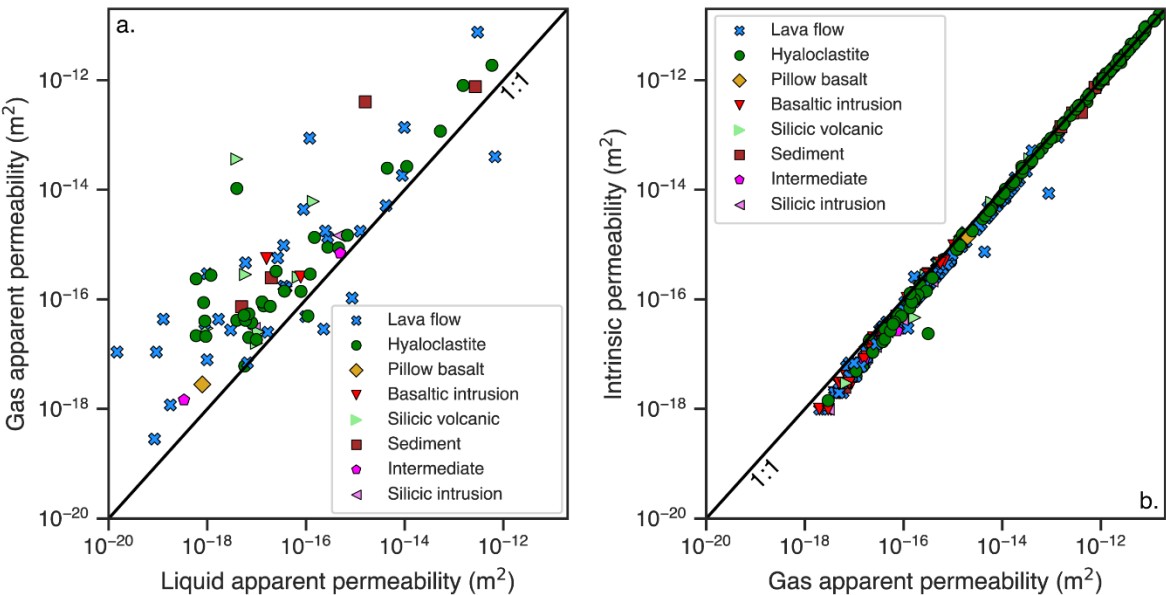

**Figure 6. Permeability data. a.** Comparison of liquid apparent permeability and gas apparent permeability. **b.** Comparison of gas apparent permeability and intrinsic permeability; the difference between these two quantities mainly reflects the Klinkenberg correction (see text).

### 3.3 Electrical Conductivity

Electrical conductivity (and its inverse, electrical resistivity) are intrinsic rock properties that measure how strong the material resists electrical current, as given by Ohm's law, $R = \frac{U}{I}$, where $R$ is resistance, $U$ voltage, and $I$ current. When an alternating current flows through a material, the electrical resistivity is characterized not only by the ratio of the magnitude of current and voltage, but also the difference in their phases, as expressed by the electrical impedance, a complex number written as a function of angular frequency $\omega$ and phase angle $\theta$ for sinusoidal current and voltage as:

$$Z(\omega) = \frac{U_o e^{i(\omega t + \theta_U)}}{I_o e^{i(\omega t + \theta_u)}} = |Z(\omega)|e^{i\theta(\omega)} = Z'(\omega) + iZ''(\omega) \qquad (8)$$

where $(U_o, \theta_U)$ and $(I_o, \theta_I)$ are the amplitudes and phases of the sinusoidal voltage and current, respectively, $t$ is time, and $\theta(\omega) = \theta_I - \theta_U$ is the frequency-dependent phase angle (generally negative, corresponding to a phase delay of voltage relative to current). For sample plugs with a given length $L$ and cross-sectional area $A$, the electrical conductivity $\sigma$ is related to $Z$ by:

$$\sigma(\omega) = |\sigma(\omega)|e^{i\theta(\omega)} = \sigma' + i\sigma'' = \frac{Z'(\omega)}{|Z(\omega)|^2}\frac{L}{A} + \frac{Z''(\omega)}{|Z(\omega)|^2}\frac{L}{A} \qquad (9)$$

where $|\sigma|$ and $\theta$ are the frequency-dependent modulus and phase angle of the complex conductivity, respectively, and $\sigma'$ and $\sigma''$ are the in-phase (real) and quadrature (imaginary) parts of the complex conductivity, respectively.

The complex conductivity is obtained by measuring the impedance spectrum of the sample over a frequency range (generally 0.1 to $10^6$ Hz). Flovenz et al. (2005) used a Zahner IM-6 electrochemical workstation, while Lévy et al. (2018, 2019a, 2019b, 2020) used a Solartron 1260 impedance meter. Results show that there is a slight dependence of resistivity on measurement frequency, particularly above 10 Hz (Flovenz et al., 2005). Two types of sample-holders and configurations were used: 1) the two-electrode set-up, where the sample is sandwiched between two metallic electrodes acting as current and voltage electrodes, and 2) a four-electrode set-up (Vinegar & Waxman, 1984), where the voltage and current electrodes are separated. In the latter, metallic (Ni, Pt, Ag) electrodes are used to inject the current and non-polarizable Ag/AgCl electrodes are used for voltage measurement. Although the four-electrode set-up improves the quality of the conductivity spectra, especially below 10 kHz, the values obtained by both set-ups are comparable at 1 kHz, where effects of electrode polarization are negligible (Lévy et al., 2019b).

In rocks where free ions in pore water are the only charge carriers, the in-phase conductivity of a volume of rock $\sigma_{bulk}$ is governed by Archie's law:

$$\sigma_{bulk} = \frac{\sigma_w}{F} \tag{10}$$

where $\sigma_w$ is the conductivity of the pore fluid and $F$ is the formation factor, representing the tortuosity of the current path. While $F$ is related to the porosity, the relationship is more indeterminate in igneous rocks than in sedimentary rocks (see e.g., Lévy et al., 2018). In rocks containing clay minerals, an additional term influences the rock conductivity, and the simplest way of writing this additional term is given by Rink and Schopper (1974):

$$\sigma_{bulk} = \frac{\sigma_w}{F} + \sigma_s \tag{11}$$

where $\sigma_s$ is the "surface" or "interface" conductivity, resulting from ion exchange with the solid matrix. More complex equations also describe the contribution from clay minerals (see e.g., Waxman and Smits, 1968 and Lévy et al., 2018) but this linear equation is often preferred. The formation factor and surface conductivity are typically determined by a series of conductivity measurements on the same sample saturated at different pore fluid salinities (conductivities). Uncertainty estimations for the formation factor can be found in Lévy et al. (2019b), including corresponding equations for this uncertainty calculation.

Figure 7 shows an example of how measurements of electrical conductivity vary as a function of the salinity of the saturating fluid and the initial smectite content of the rock. In Figure 7, sample L22 corresponds to a lava flow altered to smectite-zeolite facies alteration, and sample L48 to a lava flow altered to chlorite-epidote facies alteration. The higher smectite content in L22 results in a higher Cation Exchange Capacity (CEC, see below) compared to L48. A higher CEC corresponds to an increasing

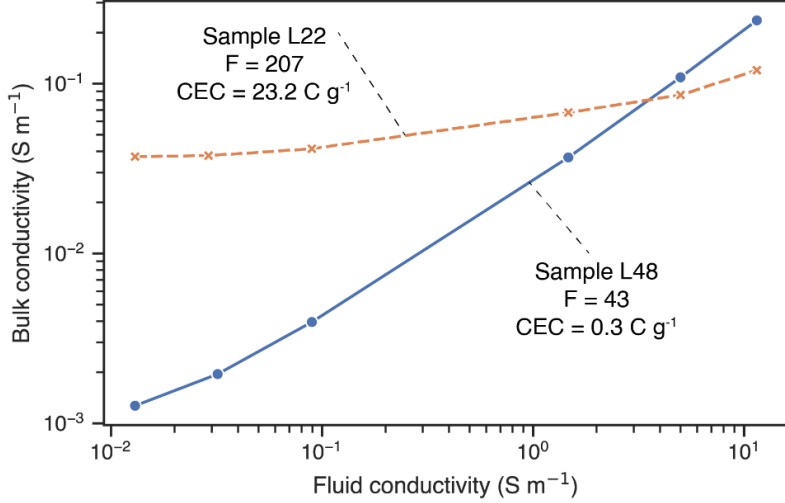

**Figure 7. Results from electrical measurements at 1 kHz versus fluid conductivity for two samples from Lévy et al. (2018) with different electrical properties. The formation factor (F) and cation exchange capacity (CEC) for the two samples are noted.**

role of surface conduction, resulting from ion exchange reactions with clays. The larger formation factor in L22 is explained by the presence of smectite alteration minerals, which partially clog the original pore network and prevent the diffusion of free ions in the pore space by the fluid, but also allow efficient conduction of electrical charge along the smectite clays.

For the samples analyzed by Franzson & Tulinius (1999), resistivity was measured at only a single fluid salinity, and an apparent formation factor $F_{app}$ was estimated by $\frac{1}{F_{app}} = \frac{\sigma_{bulk}}{\sigma_{fluid}}$. The apparent formation factor is equal or close to the true formation factor if (i) there are no clay minerals or (ii) the pore space is saturated with high-salinity fluid (the contribution from free ions in the pore space largely dominates that of clay minerals). However, when resistivity measurements are carried out on samples that may contain clay minerals and have been saturated with only one fluid salinity, care should be taken when interpreting $F_{app}$ because the contribution of surface conduction to the bulk conductivity will be significant for altered rocks, and these data points are outliers compared to the samples analyzed by Flovenz et al. (2005) and Lévy et al. (2018, 2019a, 2019b, 2020) (Fig. 12 b).

### 3.4 Petrographic and geochemical characterization

Four different methods are used to characterize the mineralogy and geochemistry of the samples: 1) Point-counting by petrographic observation, 2) XRF, 3) XRD and 4) CEC. The former two methods were used exclusively on the samples analyzed by Orkustofnun and Iceland Geosurvey (Sigurðsson and Stefánsson, 1994; Guðmundsson et al., 1995; Franzson et al., 2008; Friðleifsson & Vilmundardóttir, 1998; Franzson & Tulinius, 1999; Franzson et al., 2011). The latter two methods were used to assess the mineralogy in core samples from Lévy et al. (2018, 2019a, 2019b and 2020b).

Point counting was used to quantify primary porosity, i.e. the original open space in rock prior to alteration (dominantly vesicles and minor fractures), and to assess how much of that porosity had been filled by deposition of alteration minerals. Generally, two hundred points were counted on each rock thin section and grouped into the following categories: primary mineral, altered primary mineral, precipitate in vesicles, precipitate in fractures, intercrystalline pores, and unfilled fractures. A more detailed classification scheme was used in Friðleifsson & Vilmundardóttir (1998), with primary minerals separated by plagioclase, pyroxene, olivine, and opaque minerals. For the hyaloclastite samples investigated in Franzson et al. (2011), 1000 points were counted and grouped as one of the following: porosity, unaltered glass, altered glass, unaltered primary mineral, altered primary mineral, zeolite, clay, calcite or other.

Bulk rock chemical analyses were performed by two commercial chemical laboratories, The Caleb Brett Laboratory in England and McGill University in Canada, using standard XRF techniques (e.g., Potts and Webb, 1992; Rousseau et al., 1996). Both labs used the fused bead technique for major elements and pressed powder pellets for the determination of trace elements. Values for samples analyzed by both laboratories are generally within analytical error (Rousseau et al., 1996). The samples were analyzed for major, minor, and several trace elements (Zr, Y, Zn, Cu, Rb, Sr, Nb, Ga, Ce, V, Pb, U, Th and As). Major element analyses in the database are presented both in unnormalized form and after removing LOI (loss on ignition, see below) and renormalizing the composition to 100%. Figure 8a shows the samples categorized by lithology plotted on a total alkali – silica diagram (Le Maitre et al., 2002). Note that this diagram shows both altered and unaltered rocks. Most of the samples

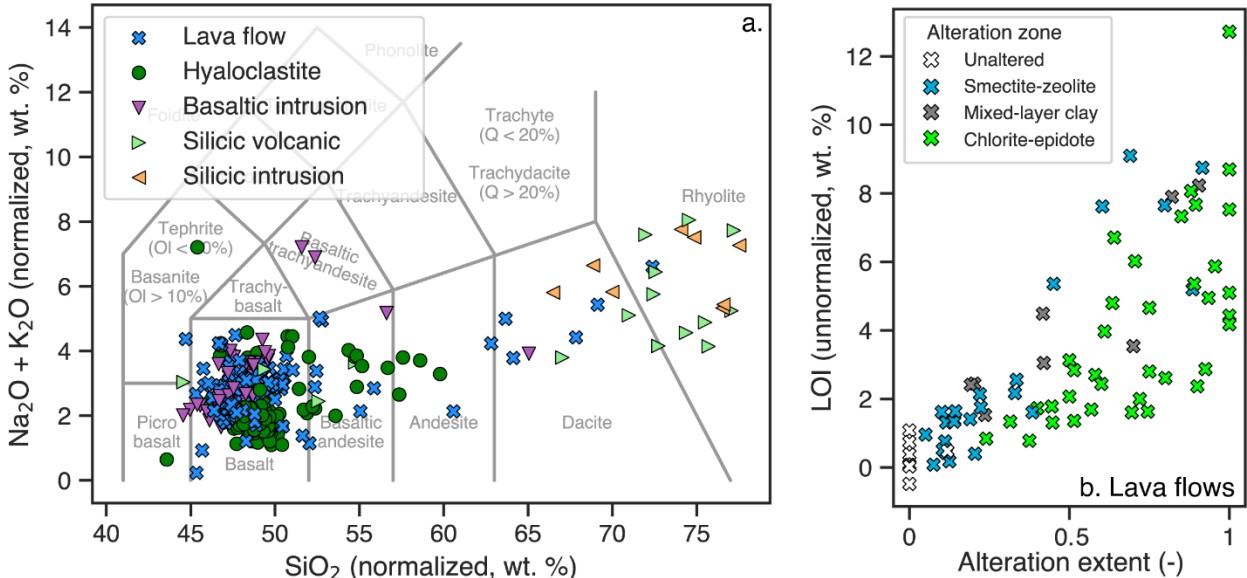

**Figure 8. Whole-rock geochemical data and petrographic observations of alteration. a. Total alkali – silica (TAS) diagram (Le Maitre et al., 2002), with samples colored by lithology. Note that many of the rocks are very altered, and that lithologic identifiers are assigned based on the interpretation of the geological context and visual characteristics, rather than the geochemical field into which the rock plots. b. Loss on ignition (LOI) versus the total extent of alteration quantified by point counting. Only lava flows are shown, and symbols are colored by alteration zone.**

plot in the basalt field, as expected. Although it has been suggested that hydrothermal alteration in Icelandic rocks is close to an isochemical process (Franzson et al., 2008), many lava flows, hyaloclastites, and basaltic intrusions plot outside of the basalt field due to the effects of intensive quartz precipitation and silicification. In addition, many of the silicic volcanic rocks or silicic intrusions appear to be depleted in silica compared to what would be expected given the lithological identifier, which is based on geologic context and visual characteristics. One chemical metric quantifying the extent alteration is loss on ignition (LOI), which was measured in many of the samples; while LOI is the sum of $H_2O$ and $CO_2$, $H_2O+$, $CO_2$ and $S_{total}$ were additionally separately analyzed in a subset of samples. Figure 8b shows the clear relationship between LOI and the extent of alteration quantified by point counting in a subset of lava flows, with the LOI increasing up to a maximum of ~12 wt. % as the degree of alteration approaches 100%.

The mineralogy of core samples from Lévy et al. (2018, 2019a, 2019b and 2020b) were analyzed using XRD. Quantitative (crystalline) phase analysis was performed using Rietveld refinements of XRD patterns on randomly oriented mounts of whole rock powder samples. The powders were front-loaded onto the sample holder, using a razor blade to smoothen the surface and avoid preferred orientations (Bish et al., 1989). When several clay minerals with overlapping peaks (e.g. smectite and mixed layer smectite-chlorite) are present, Rietveld-refinements pose a problem of ambiguity for the quantitative analyses. Therefore, smectite quantification was performed by CEC measurements, after Lévy et al. (2020a) found a linear correlation between CEC and smectite content quantified using X-ray diffraction in samples where smectite is the only clay mineral.

The CEC represents the total capacity of a rock medium to hold exchangeable cations, and is the sum of variable (pH-dependent) CEC and permanent CEC. While CEC in soil science is typically expressed in units of milli equivalent (meq), or mmol of electrons per 100 g rock, this is numerically equivalent to a given amount of charge per kg (1 meq 100 $g^{-1}$ = 965.8 Coulomb $kg^{-1}$). To measure CEC on core samples from Krafla, Lévy et al. (2020a) modified a protocol originally designed to measure CEC on pure clay samples (Meir and Kahr, 1999) that uses Copper-triethyletetramine(II) 'Cu-trien'. The smectite content was then determined using the formula:

$$\frac{CEC}{CEC_0} = \frac{\rho_{dry}}{\rho_{smec}} \qquad (12)$$

where $\rho_{dry}$ and $\rho_{smec}$ are the dry density of the sample and the density of smectite (in $g/cm^3$), respectively. The ratio $\frac{CEC}{CEC_0}$ is used as a measure of the smectite weight fraction, with $CEC_0$ = 91 meq/100 g the average CEC of pure smectite in these types of samples (Lévy et al. 2020a).

### 3.5 Acoustic velocities and mechanical properties

Acoustic velocities are available for 295 samples in the database (Franzson and Tulinius, 1999; Frolova et al., 2005; Reinsch et al., 2016; Lévy et al., 2020b). Acoustic velocities express the propagation rate of mechanical waves – compressional P-waves and shear S-waves – in a bulk environment, composed of solid minerals and fluid in pores and fractures, either gas or liquid. For the samples analyzed by Frolova et al. (2005) and Franzson et al. (2011), sonic wave velocities were measured with the ultrasonic pulse transmission technique according to Russian State Standard, 21153.7–75 (1975). The travel times for P-

waves were calculated using the time cursor on the oscillogram. Then, velocities were calculated from the core length and the travel time measurement. The frequency of the pulser was 1 MHz for dense samples and 250 kHz for more porous samples. The measurements were done in dry as well as in water-saturated states. The samples analyzed by Franzson and Tulinius (1999) were only measured in the saturated state by the Danish Geotechnical Institute, using a standard Hoek cell with lead foil between the specimen and end piston to ensure contact. The samples were loaded in a hydrostatic stress state to 1 MPa, 2.5 and 10 MPa and P-wave velocities were measured at each stress level; as the measured values show little dependence on confining stress between 1-2.5 MPa, the values presented in the database are the averages of these two measurements. The samples of Flovenz et al. (2005) were analyzed in the saturated state by Jaya et al. (2010) at the German Research Center for Geosciences (GFZ). Briefly, ultrasonic measurements were performed with piezoelectric ceramics Stelco-type P850 and PPK62, for compressional- and shear-waves, respectively. The ultrasonic excitation signal was a rectangular voltage single burst at 1.0V and 400kHz. The signal was magnified by an Amplifier Research 50A220 before being used for excitation. Though an S-wave measurement was also performed, only the recorded P-wave data is provided since the S-wave arrivals were strongly attenuated. For the samples analyzed by Reinsch et al. (2016) and Lévy et al. (2020b), seismic wave velocities were measured at both dry and saturated conditions at the University of Montpellier using coupled piezoelectric transducers with a 500 kHz resonance frequency. A transmission wave technique with a transmitter and receiver was applied, wherein an electrical spike impulsion (amplitude -400 V, width 10 ns) is sent on the transmitter, and the acoustical wave transmitted through the rock is analyzed on an oscilloscope with 200 MHz of bandwidth and a sampling frequency of 2GHz. A coupling gel was used for $V_p$ and honey for $V_s$. The relatively accuracy for the measurements is estimated to be ~3%. While the arrival time of P-waves is relatively easy to observe, that of S-waves arrival is often more ambiguous, yielding higher uncertainties. Thus, there is a greater abundance of data for P-wave velocities than S-wave velocities, as well as a greater abundance of data for dry than saturated conditions.

Mechanical parameters refer to the strength characteristics of rocks and their potential for deformation and include the Young's modulus ($E$), Poisson's ratio ($\nu$), uniaxial compressive ($\sigma_c$) and tensile ($\sigma_t$) strength. The unconfined compressive strength is measured by loading the samples and recording the pressure at failure ($P_{fail}$):

$$\sigma_c = \frac{P_{fail}}{A} \tag{13}$$

where $A$ is the sample cross-sectional area. The Young's modulus is obtained by the slope of the stress-strain curve, usually at a point of 50% of $\sigma_c$, according to:

$$E = \frac{\partial \sigma}{\partial \varepsilon} \tag{14}$$

Poisson's ratio is the ratio between the radial- and axial-strain defined as:

$$\nu = \frac{\partial \varepsilon_{radial}}{\partial \varepsilon_{axial}} \tag{15}$$

This was only measured for a few samples subject to triaxial tests. The tensile strength was only measured by Árngrimsson and Gunnarsson (1999) using the so-called Brazilian disk method, where a circular cylindrical sample is compressed along its diameter and strain is measured. The tensile strength (in MPa) was calculated using the equation:

$$\sigma_t = 0.636 \frac{P_{fail}}{Dt} \qquad (16)$$

where $D$ is the sample diameter (mm) and $t$ the thickness of the sample (mm).

For the hyaloclastites samples analyzed by Frolova et al. (2005) and Franzson et al. (2011), the uniaxial compressive strength test was performed by standard testing procedures in accordance with Russian State Standard 21153.2-84 (1984) and ASTM D7012 (2013). Uniaxial compressive strength was measured using a German hydraulic press CDM-10/91, and was determined for samples in dry and water-saturated states. The samples analyzed by Árngrimsson and Gunnarsson (1999) were analyzed at

the Technical University of Denmark (DTU), which performed triaxial tests on five samples, and the Danish Geotechnical Institute (GEO), which performed Brazilian disk tests on 55 samples and unconfined compressive strength tests on 36 samples, with methods according to ISRM standard (Ulusay and Hudson, 2007). The elastic constants given for the samples from Frolova et al. (2005), Franzson and Tulinius (1999) and Jaya et al. (2010) were calculated from the measured wave velocities and the bulk density (e.g., Mavko et al., 2009).

**3.6 Thermal properties**

Thermal conductivity measurements for two samples described in Franzson and Tulinius (1999) were carried out by the Department for Geophysics at the University of Aarhus. The thermal conductivity was measured under saturated conditions using the divided bar technique and the vertical stress level was 1 MPa. In addition, thermal conductivity measurements were performed for the samples analyzed by Friðleifsson & Vilmundardóttir (1998) and Guðlaugsson (2000) by New England

Research (Johnson and Boitnott, 1998). For the latter study, 57 measurements of thermal conductivity under unsaturated conditions were performed on samples obtained from a single unaltered lava flow in the Reykjavik area.

**4 Data availability**

The database is archived at Zenodo at https://doi.org/10.5281/zenodo.6980231 (Scott et al., 2022b) and is available under the Creative Commons Attribution 4.0 International license. This repository includes the main Excel file containing the two main

worksheets: one listing sample metadata and petrophysical properties, a second listing geochemical and petrographic data. In addition, we provide a zip file containing photographs of the sample sites, and an Excel file listing the file names of photographs corresponding to the sample IDs All worksheets are additionally included in the repository as csv files with the separator '|'. The photographs are also included in the Zenodo repository as a separate directory.

## 5 Results and Discussion

Although most of the roughly ~21000 datapoints contained in Valgarður have presented in previous studies (Table 1), combining the data from different types of studies (petrophysical, geochemical, petrographical) can elucidate the relationship between lithology, alteration, physical properties and rock chemical and mineralogical composition. Full description of this relationship is beyond the scope of this article. Here, we provide summary statistics for porosity, grain density and permeability, describe how lithology and interconnected porosity control the alteration process, and show how alteration

controls physical properties such as porosity, permeability, grain density, resistivity, elastic wave velocities and strength.

### 5.1 Summary statistics

The roughly ~21000 datapoints comprising the database reveal complex relationships between lithology, alteration, physical parameters and chemical/mineralogical composition. Table 5 provides summary statistics by rock type and alteration zone for some of the best-characterized parameters (connected porosity, grain density, and intrinsic permeability). These statistics

highlight some expected trends – such as the higher porosity of hyaloclastites (0.2-0.5) compared to lava flows (0.01-0.35) and basaltic intrusions (0.01-0.15), the decrease in porosity that generally accompanies alteration, and the lower grain density of silicic rocks (~2.5 g cm$^{-3}$) compared to basaltic rocks (2.8-3 g cm$^{-3}$ for basaltic lava flows and intrusions). Notably, Table 5 indicates that smectite-zeolite altered hyaloclastites tend to have high porosity (0.2-0.3), low grain density (~2.7-2.8 g cm$^{-3}$) and relatively high permeability (~10$^{-14}$ m$^2$). Hyaloclastites altered to mixed-layer clay, chlorite-epidote, or chlorite-epidote

conditions, which tend to develop at hotter temperature conditions (≳250 ºC) in active geothermal systems, show lower permeability (~10$^{-16}$ m$^2$) but maintain relatively high porosity (≳0.2). Basaltic intrusions tend to show alteration at high-temperature conditions along with lower porosity (0.02-0.1), higher grain density (2.9-3 g cm$^{-3}$) and lower permeability (~10$^{-17}$ m$^2$).

**Table 5. Summary statistics for connected porosity, grain density and intrinsic permeability, for samples categorized by broad lithological identifier and alteration zone. The number of samples used to calculate the mean and standard deviation is presented in parentheses. Data is only presented for lithology – alteration zone combinations with sufficient data to calculate summary statistics. Note that the values given for intrinsic permeability were calculated using the decadic logarithm of the data.**

| Lithological category | Alteration zone | Connected porosity (-) | Grain density (g cm$^{-3}$) | Intrinsic permeability (decadic logarithm, m$^2$) |
|---|---|---|---|---|
| Lava flow | Unaltered | 0.13 ± 0.10 (267) | 3.03 ± 0.07 (267) | -15.2 ±1.06 (97) |
| | Smectite-zeolite | 0.09 ± 0.10 (50) | 2.87 ± 0.09 (50) | -16.8 ± 1.10 (25) |
| | Mixed-layer clay | 0.10 ± 0.08 (40) | 2.86 ± 0.09 (37) | -17.1 ± 0.91 (27) |
| | Chlorite-epidote | 0.06 ± 0.07 (57) | 2.86 ± 0.10 (57) | -16.5 ± 1.15 (19) |
| | Epidote-actinolite | 0.06 ± 0.06 (38) | 2.84 ± 0.10 (38) | -16.8 ± 0.87 (30) |
| Hyaloclastite | Unaltered | 0.34 ± 0.10 (114) | 2.73 ± 0.17 (113) | -12.9 ± 1.7 (82) |
| | Smectite-zeolite | 0.28 ± 0.10 (68) | 2.62 ± 0.16 (68) | -13.8 ± 1.85 (50) |
| | Mixed-layer clay | 0.24 ± 0.06 (30) | 2.71 ± 0.13 | -16.3 ± 0.60 (14) |
| | Chlorite-epidote | 0.22 ± 0.09 (40) | 2.78 ± 0.17 (41) | -16.1 ± 1.15 (11) |
| | Epidote-actinolite | 0.10 ± 0.07 (4) | 2.88 ± 0.15 (4) | -15.7 ± 0.64 (4) |
| Basaltic intrusion | Unaltered | 0.08 ± 0.06 (12) | 2.97 ± 0.10 (12) | - |

| | Smectite-zeolite | 0.04 ± 0.03 (13) | 2.89 ± 0.11 (13) | -17.4 ± 0.70 (10) |
|---|---|---|---|---|
| | Mixed-layer clay | 0.08 ± 0.05 (13) | 2.87 ± 0.08 (13) | -17.6 ± 0.97 (11) |
| | Chlorite-epidote | 0.02 ± 0.03 (35) | 2.88 ± 0.11 (35) | -16.7 ± 0.90 (18) |
| | Epidote-actinolite | 0.03 ± 0.03 (57) | 2.94 ± 0.09 (57) | -17.0 ± 0.81 (23) |
| Silicic intrusion | Unaltered | 0.12 ± 0.04 (5) | 2.54 ± 0.07 (5) | -13.0 (1) |
| | Smectite-zeolite | 0.18 ± 0.02 (3) | 2.34 ± 0.23 (3) | -17.1 ± 0.32 (3) |
| | Mixed-layer clay | - | - | - |
| | Chlorite-epidote | 0.08 ± 0.09 (6) | 2.57 ± 0.13 (6) | -16.1 ± 0.87 (5) |
| | Epidote-actinolite | 0.09 ± 0.04 (10) | 2.71 ± 0.10 (10) | -16.8 ± 0.84 |
| Silicic volcanic | Unaltered | 0.11 ± 0.11 (36) | 2.50 ± 0.06 (36) | -13.0 (1) |
| | Smectite-zeolite | 0.13 ± 0.08 (24) | 2.51 ± 0.20 (24) | -16.4 ± 1.11 (21) |
| | Mixed-layer clay | - | - | - |
| | Chlorite-epidote | 0.14 ± 0.07 (7) | 2.7 ± 0.1 (7) | -16.3 ± 0.44 (5) |
| | Epidote-actinolite | 0.07 (1) | 2.67 (1) | -17.16 (1) |

Note that the permeability values presented in Table 4 are representative of matrix permeability, and permeability measured in the field (e.g., using well testing) commonly exceeds the matrix permeability by an order of magnitude or more due to the control of fracture permeability on bulk permeability (e.g., Björnsson and Bödvarsson, 1990). Therefore, upscaling this data for reservoir modeling will require consideration of the impact of fractures on the permeability. This can be treated in an idealized sense using dual porosity/permeability models (**cite**), or by explicitly representing the fractures in models using discrete fracture networks (**cite**). Moreover, geostatistical approaches to upscaling laboratory permeability to field-scale numerical grids require knowledge of the uncertainty of matrix permeability; Table 4 indicates that the standard deviation of permeability for a given lithology-alteration zone combination often exceeds 1 order of magnitude, and the range in measured permeability for a given rock type (e.g., unaltered lava flows and hyaloclastites) can exceed 6 orders of magnitude ($10^{-17} - 10^{-11}$ m$^2$). Table 4 additionally indicates that there is a paucity of data for certain rock types, such as unaltered basaltic intrusions and silicic intrusions.

**5.2 The lithologic control on pore connectivity and alteration**

Whether alteration leads to porosity creation or destruction depends on both the type and extent of alteration as well as the primary connected porosity of the rock (e.g., Mordensky et al., 2018; Villeneuve et al., 2019). Different volcanic rocks lithologies show variability in pore connectivity linked to the geometry of the pore network formed during magma crystallization, vesiculation, fragmentation, and densification (e.g., Blower, 2001; Bernard et al., 2007; Yokoyama and Takeuchi, 2009; Wright et al., 2009; Kennedy et al., 2010; Heap et al., 2014; Colombier et al., 2017). Figure 9 shows calculated pore connectivity ($C = \frac{\phi_{connected}}{\phi_{total}}$; Colombier et al., 2017) as a function of total porosity for the samples in the database. While hyaloclastites with a porosity greater than 0.1 generally show high connectivity (>0.9), many lava flows, pillow basalts, basaltic intrusions and silicic volcanic rocks with porosity <0.1-0.3 have intermediate connectivity (0.3-0.8). Lava flows and basaltic intrusions with total porosity <0.05 show the lowest connectivity (<0.25). Note that a few samples have non-physical values

of pore connectivity >1, which is likely a result of too-low measured grain density (Colombier et al., 2017) used in the calculation of total porosity according to equation (3).

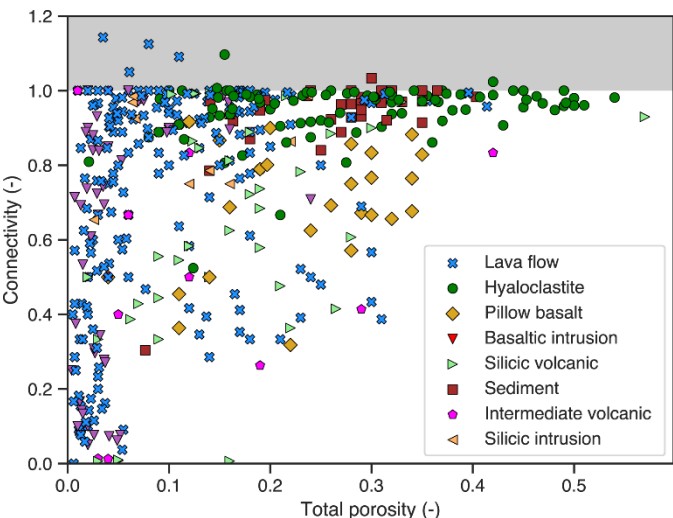

**Figure 9. Relationship of pore connectivity to total porosity. Samples categorized by lithology. The grey fields corresponds to unphysical values of connectivity (C > 1), which could result from variability in grain density (see text).**

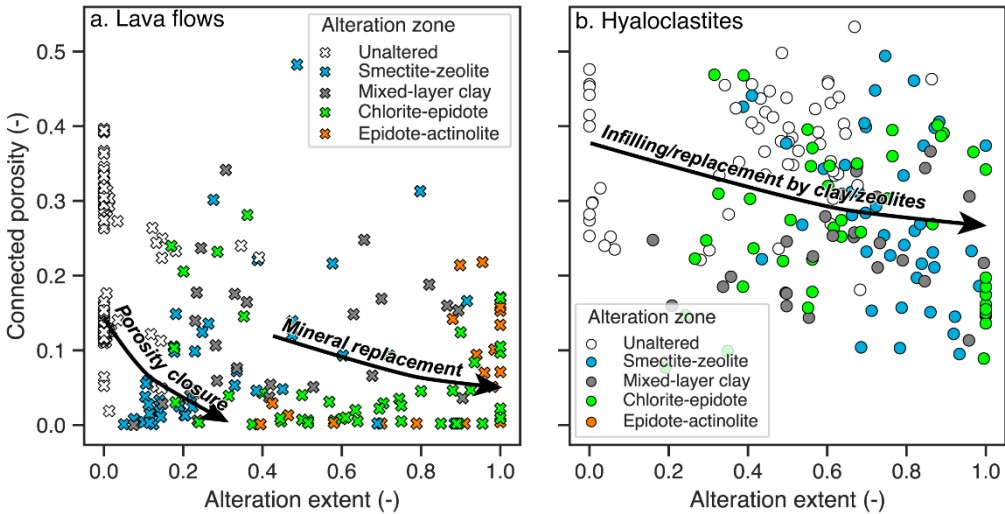

**Figure 10. Relationship between alteration extent, as determined by petrographic observations and XRD, and the connected porosity for a. lava flows and b. hyaloclastites. High porosity rocks such as hyaloclastites undergo near-complete mineral replacement in the smectite-zeolite facies, whereas rocks with low porosity, such as lava flows, undergo porosity closure after a limited extent of alteration. Rocks altered to chlorite-epidote or epidote-actinolite facies show higher alteration extents because of replacement of primary minerals and secondary minerals.**

580 Mineralogical observations derived from petrography and XRD suggest that rocks with greater connected porosity undergo a greater extent of hydrothermal alteration. Figure 10 shows this by plotting total alteration on the x-axis against connected porosity. Lava flows (Fig. 10a), which may have very high connected porosity within the vesicular margins but more usually have low connected porosity (<0.2), only undergo a relatively small extent of smectite-zeolite alteration before porosity closure limits. In contrast, hyaloclastites (Fig. 10b), which tend to have higher connected porosity (>0.2), undergo near-complete

585 replacement of glass and primary minerals at smectite-zeolite alteration. While alteration of hyaloclastites at smectite-zeolite conditions proceeds both by palagonitization and replacement of basaltic glass by smectite clay as well as infilling of clay and zeolites in vesicles, at higher temperature chlorite-epidote (>240 ºC) or epidote-actinolite (>280 ºC) conditions, alteration proceeds to an increasing extent by replacement of smectite clays formed at lower temperature conditions by chlorite as well as replacement of primary minerals to form secondary minerals such as epidote, wairakite and actinolite (e.g., Franzson and

590 Gunnlaugsson, 2021). As a result of enhanced replacement of primary minerals during higher temperature alteration, lava flows with relatively low connected porosity altered to chlorite-epidote or epidote-actinolite conditions tend to show a higher extent of alteration (>0.5). This data is consistent with geochemical modeling suggesting that rocks with a relatively low interconnected primary porosity (<0.2), such as lava flows, will tend to undergo rapid porosity closure during the volumetric changes associated with hydrothermal alteration, while rocks with a significant fraction of interconnected primary porosity

595 (>0.2), such as hyaloclastites, will undergo near-complete replacement of basaltic glass and primary minerals by secondary minerals (Thien et al., 2015).

## 5.3 Permeability-porosity relationships in altered basalts

While many previous studies have found that permeability of volcanic rocks tends to increase with increasing porosity (e.g., Saar and Manga, 1999; Blower, 2001; Farquharson et al., 2015; Wadsworth et al., 2016; Colombier et al., 2017), studies have also shown that the permeability of unaltered and altered volcanic rocks can be quite variable (e.g., Heap et al. 2017a; Mordensky et al., 2018; Villeneuve et al., 2019). Figure 11 shows connected porosity and permeability in lava flows (Fig. 11a), hyaloclastites and pillow basalts (Fig. 11b) and other lithologies (Fig. 11c). While permeability generally increases with porosity, this relationship is not very strong as there is more than five orders of magnitude variability at a given porosity. For example, unaltered lava flows show a large range in permeability ($10^{-17} - 10^{-11}$ m$^2$). The two dense clusters of data points in Fig. 11a originate from detailed investigations of a single unaltered olivine tholeiite lava flow of Pleistocene age located on Öskuhlíð, in the Reykjavik area (Friðleifsson and Vilmundardóttir, 1998; Guðlaugsson, 2000; Franzson et al., 2001). One of these clusters consists of samples obtained from the coarser, inner part of the flow and shows high permeability at relatively low connected porosity; a second cluster consists of samples originating in the more glass-rich vesicular outer margin and shows lower permeability at higher connected porosity (0.2-0.4) and. This enigmatic permeability-porosity relationship has been interpreted as the result of high pore interconnectivity in the compact flow interior due to grain boundary microcracks (the presence of which is inferred from the low P-wave velocities of these samples), whereas larger vesicles on the outer margin are largely isolated (Guðlaugsson, 2000; Franzson et al., 2001). The Öskuhlíð lava flow is relatively young (<2.5 Ma) and has not yet undergone burial. Unaltered lava flows collected from the brecciated margins of lava flows in other settings show high connected porosity as well permeabilities as high as $10^{-11}$ m$^2$.

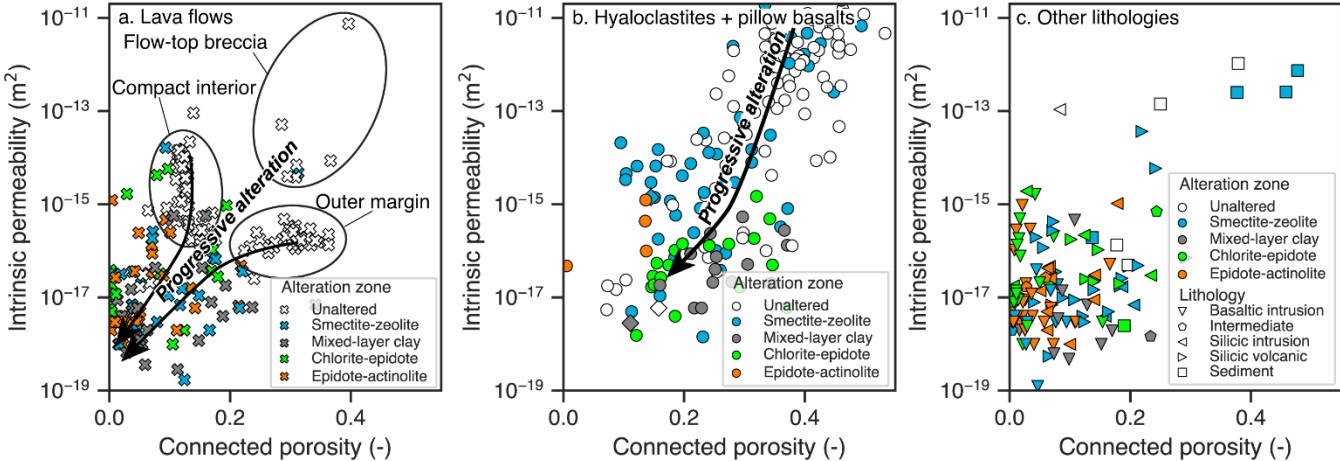

**Figure 11. Relationship between connected porosity and intrinsic permeability in a) lava flows, b) hyaloclastites and pillow basalts, and c) other lithologies, including basaltic intrusions, silicic intrusions, silicic volcanics, intermediate rocks, and sediments. Samples colored by alteration zone, with symbols corresponding to different lithology.**

Alteration generally causes permeability to decrease. However, many hyaloclastite tuffs and some lava flows with high initial connected porosity (<0.3) maintain relatively high permeability ($\gtrsim 10^{-16}$ m$^2$) even if altered at high-temperature chlorite-epidote or epidote-actinolite conditions. While the permeability of unaltered hyaloclastites and hyaloclastites altered at smectite-zeolite conditions can be as high as $10^{-11}$ m$^2$, the permeability of hyaloclastites altered at higher-temperature conditions show highly variable but lower permeability ($10^{-17} - 10^{-14}$ m$^2$) (Fig. 11b). In lava flows (Fig. 11a), progressive alteration tends to reduce the connected porosity and permeability to <0.2 and $10^{-18} - 10^{-16}$ m$^2$, respectively, some lava flows altered to chlorite-epidote or epidote-actinolite conditions have moderate permeability ($\sim 10^{-15}$ m$^2$) even though connected porosity is <0.1. Due to the relatively low connected porosity of basaltic intrusions, most of which are altered to chlorite-epidote or epidote-actinolite conditions, permeability tends to be low ($10^{-18} - 10^{-16}$ m$^2$) (Fig. 11c). Although the data is limited, the data suggest that the permeability of silicic intrusions is slightly higher ($10^{-17} - 10^{-15}$ m$^2$) compared to basaltic intrusions and the permeability of silicic volcanics can be variable ($10^{-18} - 10^{-15}$ m$^2$).

Putting this data in the context of the petrographic observations described above (Fig. 10), high porosity, glass-rich hyaloclastites maintain high permeability during the alteration process because of their high primary porosity, thereby allowing greater fluid through-flux and facilitating progressive alteration (basaltic glass dissolution and secondary mineral precipitation). More rapid porosity closure in response to low-grade alteration in lava flows results in more rapid permeability decrease, which limits the alteration extent. Although the time-temperature-fluid flux conditions experienced by different rocks is highly variable, the data suggests that matrix permeability of high temperature chlorite-epidote or epidote-actinolite conditions can be as high as $\sim 10^{-15}$ m$^2$. The ability for rocks to maintain high permeability throughout the alteration process is of crucial importance for successful carbon mineralization in basaltic rocks. This data suggests that the precipitation of volumetrically significant quantities of carbonates and other alteration minerals (e.g. amorphous silica, clays and zeolites; Gysi, 2017) in the subsurface during the interaction of carbonated water with basaltic rock need not result in porosity closure, as long as the porosity of the target reservoir is $\gtrsim 0.25$.

## 5.4 Relationships between grain density and electrical properties in altered volcanic rocks

The database highlights physical relationships between parameters such as grain density and electrical properties and the overall control exercised by lithology and alteration. Electromagnetic methods are commonly used for geophysical exploration, as smectite-rich rocks constituting the low-permeability cap-rock in geothermal systems (Cumming, 2016) are known to have a lower resistivity (<10 Ωm) than rocks hosting the underlying high-temperature geothermal reservoir, which often shows resistivities >100 Ωm (e.g., Árnason et al., 2010; Muñoz, 2014). Figure 12a compares grain density and bulk resistivity and shows that there is generally a positive correlation between the two parameters, albeit with significant scatter. The measured data reproduces these expected trends, with the samples altered to smectite-zeolite conditions showing lower resistivity than samples altered to chlorite-epidote or epidote-actinolite conditions. Although the measurements shown in Figure 12a were performed at room temperature and using a saturating fluid of low salinity/conductivity, the effect of higher saturating fluid salinity/conductivity could be factored in through consideration of the formation factor according to equation 11. Figure 12b

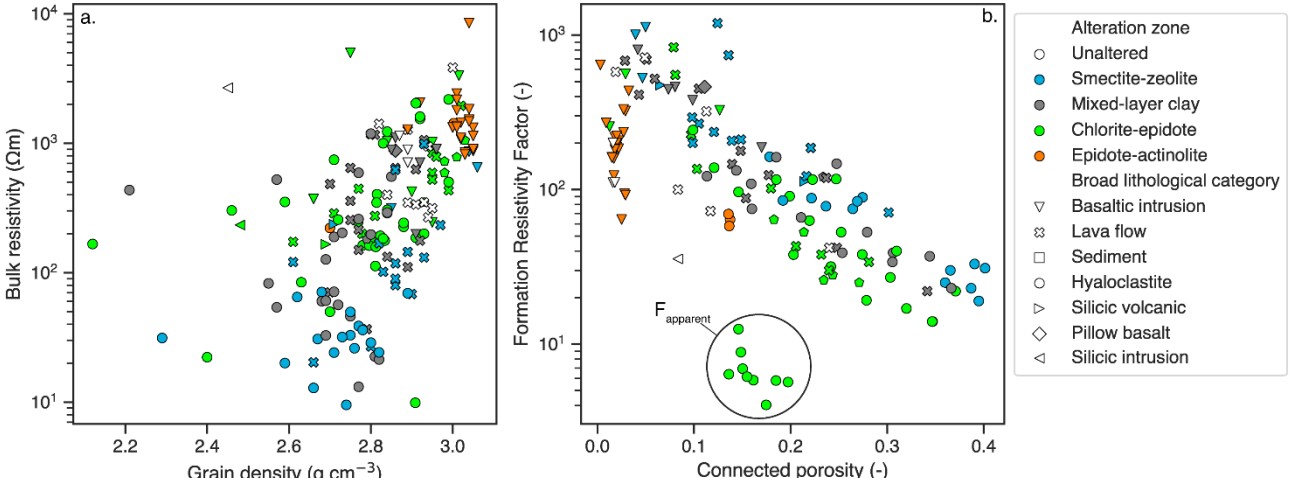

**Figure 12. a) Relationship between grain density and bulk resistivity. Measurements were performed using saturating fluid with a low salinity and conductivity (see text). b) Relationship between formation resistivity factor and connected porosity. Note that the outliers used the apparent formation resistivity factor (see text).**

shows that the calculated formation resistivity factor shows a clear log-linear relationship to connected porosity, with smectite-

rich rocks having a higher formation factor at a given porosity. Note that the outlier data points with a low formation resistivity

factor <10 in Fig. 12b all originate from Franzson and Tulinius (1999), which calculated the apparent formation resistivity

factor, i.e., where conductivity measurements were not performed with saturating fluid of variable salinity/conductivity.

The lower resistivity of smectite-rich rocks has long been known both based on field measurements described above as well

as experimental studies (e.g., Flovenz et al., 1985). However, Fig. 12a points to a physical relationship between grain density

and resistivity in geothermal reservoir rocks. Although the grain density of smectite can range from 2-2.6 g cm$^{-3}$ depending on

its hydration status, this is appreciably lower than that of chlorite (2.6-3.3 g cm$^{-3}$; Deer et al., 1996). While the relationship

between low grain density and high conductivity has been seen in other experimental studies (e.g., Nelson and Anderson,

1992), this is perhaps underappreciated in the context of geophysical imaging in geothermal systems. This data suggests that

joint inversions of gravimetric and electromagnetic measurements, particularly when combined with porosity constraints based

on core measurements and geologic modeling (e.g., Soyer et al., 2018), may facilitate better delineation of the geometry of the

cap rock, precise knowledge of which is essential to target the underlying higher temperature resistive core.

### 5.5 Elastic wave velocities and mechanical properties

Figure 13a shows that compressional (P-wave) velocities are inversely correlated to porosity: basaltic intrusions show the

highest velocities and the lowest porosities, while hyaloclastites have the lowest velocities and higher porosities. This

relationship has been seen in several previous studies of volcanic rocks (e.g., Pola et al., 2014; Frolova et al., 2014; Wyering

et al., 2014; Heap et al., 2015; Durán et al., 2019; Frolova et al., 2021). However, the data show considerable scatter, and the

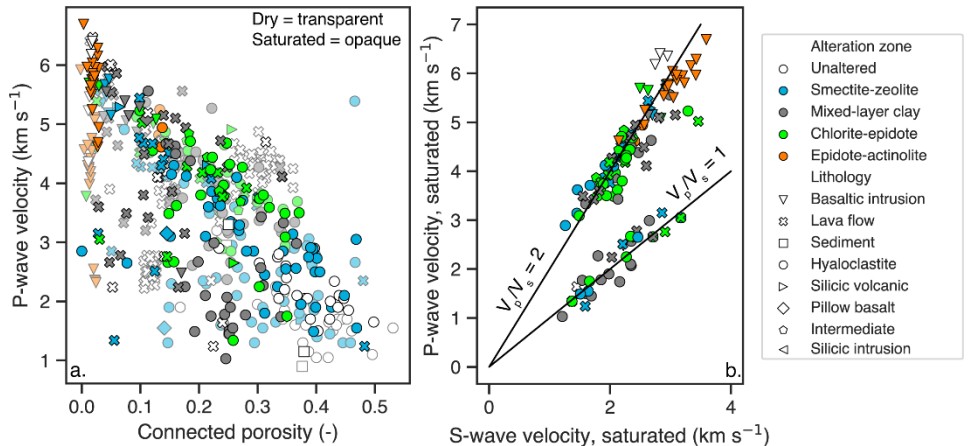

**Figure 13. a. Compressional (P-wave) velocities under dry (transparent symbols) and saturated (opaque symbols) conditions versus connected porosity. b. S-wave velocities versus P-wave velocities under saturated conditions.**

relationship between acoustics velocities measured at dry conditions (shown by transparent markers) and saturated conditions (shown with opaque markers) is complex (e.g., Nur and Simmons, 1969; Kahraman, 2007; Kahraman et al., 2017). Most of the samples show P-waves that tend to travel faster in a water-saturated than dry environment, but several hyaloclastites and

lava flows show dry velocities that are systematically ~1-2 km s$^{-1}$ greater than the saturated velocities. Figure 13b compares P-wave velocity with the S-wave velocity at saturated conditions. Most of the samples with saturated P-wave velocities greater than ~2 km s$^{-1}$ show $V_p/V_s$ close to ~2. However, many of the samples with low P-wave velocities (<3 km s$^{-1}$) have $V_p/V_s$ close to ~1. The latter samples have variable connected porosity but tend to be brecciated hyaloclastites or flow-top breccias. As P- and S-wave velocities are strongly dependent on crack density and geometry, highly cracked rocks may display in some cases

very low velocities at room conditions (e.g., Nur and Simmons, 1969; Vinciguerra et al., 2005; Nara et al., 2011). This could explain the low P-wave velocities seen in lava flows and hyaloclastites in Figure 13.

Alteration impacts rock strength and thereby exerts an influence on rock mechanical behavior and failure mode (e.g., Pola et al., 2014; Heap and Violay, 2021). Depending on the porosity changes during hydrothermal alteration and the abundance and type of clay minerals, alteration can increase or decrease rock strength (e.g., Wyering et al., 2014; Frolova et al., 2014; Pola et

al., 2014; Mordensky et al., 2018; Farquharson et al., 2019; Heap et al., 2020a; Frolova et al., 2021). Figure 14 shows that uniaxial compressive strength (UCS) (Fig. 14a) and Young's modulus (Fig. 14b) decrease with increasing porosity, as has been observed in several previous studies of volcanic rocks (e.g., Al-Harthi et al., 1999; Pola et al., 2014; Wyering et al., 2014; Heap et al., 2014; Schaefer et al., 2015; Mordensky et al., 2018; Coats et al., 2018; Harnett et al., 2019). However, also consistent with these studies, the data reveal significant scatter; for example, at a porosity of 0.2, UCS can range from ~10

MPa to ~100 MPa. Heap and Violay (2021) describe how such variability in rock strength can result from variable hydrothermal alteration and the partitioning of porosity between pores and microcracks and their geometrical properties.

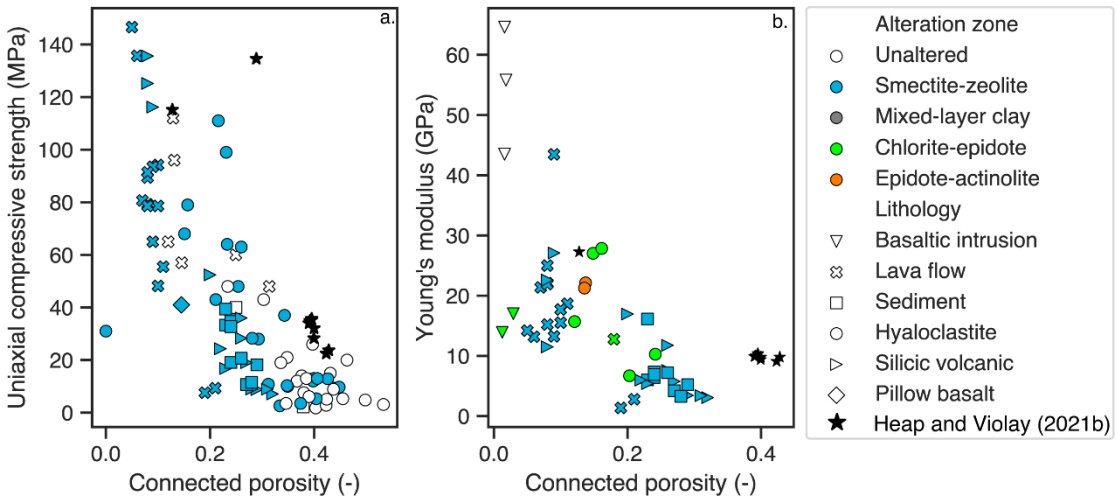

**Figure 14. a. Uniaxial compressive strength (UCS) and b. Young's modulus as a function of connected porosity. Data for Icelandic rocks from Heap and Violay (2021) shown with black stars.**

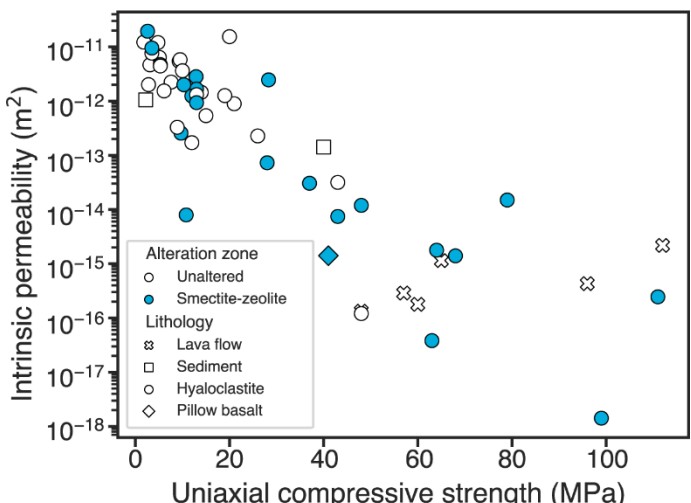

**Figure 15. Permeability as a function of uniaxial compressive strength. Note that data is only available for unaltered rocks or samples that have been altered to smectite-zeolite conditions.**

Although altered rocks can show wide variability in strength and permeability, the inverse relationship between rock strength and permeability, as shown in Figure 15, results from the greater presence of interconnected porosity and microcracks in relatively weak rocks. On the field-scale, the low permeability of the smectite-rich cap-rock is a result of reduced fracture cohesion and strength, which limits shear and inhibits dilatation of fractures (Dobson et al., 2003; Neuzil, 1994; Davatzes and Hickman, 2010; Wyering et al., 2014; Meller and Kohl, 2014; Sanchez-Alfaro et al., 2016) and promotes ductility (Mordensky et al., 2019). In contrast, rocks that have been altered to propylitic conditions and contain a greater abundance of secondary minerals such as chlorite, quartz and epidote retain rock strength in response to alteration and consequently show brittle dilatant

behavior during slip (Davatzes and Hickman, 2010; Meller and Kohl, 2014; Sanchez-Alfaro et al., 2016). As field-scale
permeability in geothermal systems is commonly fracture-controlled (e.g., Lamur et al., 2017; Heap and Kennedy, 2015; Farquharson and Wadsworth, 2018; Jolie et al., 2021), increased rock strength may facilitate the development of fracture permeability needed to result in productive geothermal wells (e.g., Villeneuve et al., 2019). However, Figures 14 and 15 show that at present there is very limited data for Icelandic rocks altered to chlorite-epidote and epidote-actinolite conditions.

### 5.6 Thermal conductivity

Previous studies of Hawaiian basalts have shown that thermal conductivity decreases with increasing porosity and increases if the samples are saturated with water (Robertson and Peck, 1974). Although the thermal conductivity data in this study is mainly limited to samples derived a single unaltered lava flow in the Reykjavik area (Guðlaugsson, 2000), the data suggest a similar relationship (Figure 16). Thermal conductivity measured at unsaturated conditions ranges from ~1-2 W m$^{-1}$ K$^{-1}$, with a general trend suggesting increasing thermal conductivity at lower connected porosity. In contrast, thermal conductivity
measured under saturated conditions on two hyaloclastite samples obtained from the ÖJ-1 borehole is significantly higher, ranging from 2.5-2.75 W m$^{-1}$ K$^{-1}$. Although at present there is insufficient data to characterize the effect of lithology and alteration zone on thermal conductivity, Figure 16 indicates that significant variability in thermal conductivity within a single lithological unit results from the heterogenous distribution of pore space.

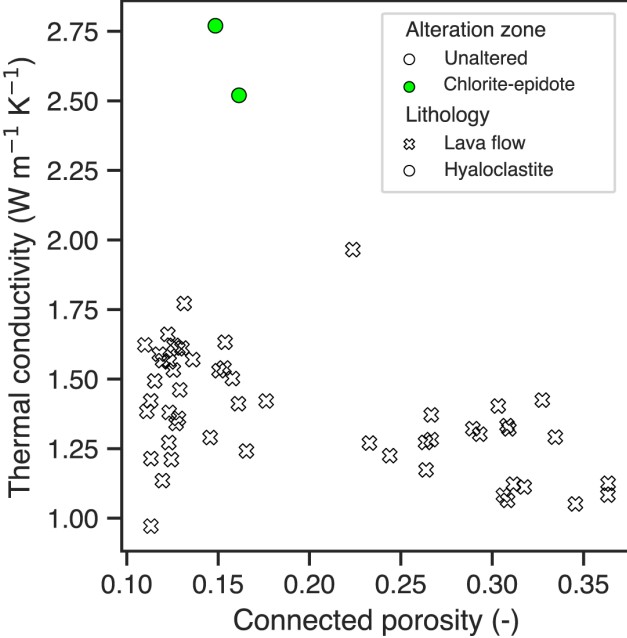

**Figure 16. Thermal conductivity as a function of connected porosity. Note that most of the available data is derived from a single unaltered lava flow (Guðlaugsson, 2000). Measurements on the lava flows were performed at unsaturated conditions, whereas measurements on the hyaloclastite samples were performed under water-saturated conditions.**

## 6 Concluding remarks, limitations and future status of the database

The efforts of geologists in Iceland over the past 50+ years have resulted in a tremendous amount of data spanning petrophysical, geochemical, and petrographic analyses of a wide range of basaltic rocks and associated silicic and intermediate rocks. However, it has historically been common that after publication of a paper or report, the only remaining manifestations of the data are the figures contained in the publication, and much (or all) of the raw underlying data was inaccessible. This practice is only starting to change in response to increasing emphasis on data availability. The motivation of the *Valgarður*

database is to ensure that the data resulting from decades of intensive study of Icelandic geothermal reservoir rocks remain accessible to future generations of geoscientists.

   In this paper, we have described the methods used to acquire the data in some detail, and briefly characterized some aspects of the relationship between lithology, alteration, and petrophysical properties. The different lithologies show systematic differences in interconnected porosity, with lava flows and basaltic intrusions showing lower interconnected porosity (<0.2)

than hyaloclastites (>0.3). We propose that a result of their higher interconnected porosity, hyaloclastites can undergo near-complete replacement of glass and primary minerals during smectite-zeolite alteration and maintain relatively high interconnected porosity (>0.2), while smectite-zeolite alteration in lava flows and intrusions is limited by porosity closure. By reducing interconnected porosity, alteration exerts a first-order control on petrophysical properties such as porosity, permeability, grain density, resistivity, acoustic velocities, and strength. Generally, altered rocks are characterized by lower

porosity, lower permeability, lower grain density, lower resistivity and increased strength. However, there is significant variability depending on lithology and the type/extent of hydrothermal alteration. Although full description of the relationship between lithology, hydrothermal alteration and rock chemical composition is beyond the scope of this article, we believe that the database would be well-suited for such a study, as many of the samples are highly altered and the database contains whole-rock geochemical data for ~350 samples.

In the database, basaltic rocks are better characterized than silicic rocks, and there is a relative paucity of mechanical and thermal data, particularly for rocks altered at high-temperature (>240 ºC) chlorite-epidote or epidote-actinolite conditions. Although at present we restrict the database to measurements at near-ambient conditions, the electrical, mechanical and transport properties of Icelandic rocks have also been measured under experimental conditions aimed at reproducing *in-situ* conditions in the subsurface (e.g., Vinciguerra et al., 2005; Jaya et al., 2010; Kristínsdóttir et al., 2010; Milsch et al., 2010;

Adelinet et al., 2010; Adelinet et al., 2013; Grab et al., 2015; Eggertsson et al., 2020a,b; Nono et al., 2020; Kummerow et al., 2020; Weaver et al., 2020). In the future, it would be sensible to include measurements performed at variable pressure, temperature and saturating fluid salinity into the database. However, as most of the measurements included in this database (Table 1) were only performed at near-ambient conditions, we felt that inclusion of this additional data measured at variable conditions would overly complicate the structure of the database and thereby ultimately limit its usability.

Although the database is restricted to Iceland, we believe that the data contained in this database provides useful constraints on the petrophysical properties of basaltic rocks outside of Iceland. Similar relationships between alteration type/extent and

petrophysical properties have been observed in previous studies performed using altered volcanic rocks obtained from geothermal systems in New Zealand (Heap et al. 2017a; Mordensky et al., 2018; Villeneuve et al., 2019) and Mexico (Weydt et al., 2022). However, the limitations of the database should also be acknowledged. For example, thermal conductivity measurements are only available for a relatively small number of samples, most of which were derived from a single lava flow in the Reykjavik area (Guðlaugsson, 2000). Other studies have measured the thermal properties of Icelandic rocks (Ruether, 2011), and thermal conductivity and thermal diffusivity was measured on a large number of samples obtained from a nearly 2 km long core in the Reyðarfjörður region (Oxburgh and Agrell, 1982; Drury, 1985; Flovenz and Saemundsson, 1985). To the authors best knowledge, the data obtained in these studies does not exist in tabulated form accessible over the internet. This highlights the challenges of ensuring the accessibility of such "legacy" data. As future studies address these gaps in the database, we anticipate future releases of the database using the versioning system available on Zenodo, making these updates available at the same repository address given in this publication (Scott et al., 2022a). Despite the gaps in the data, we believe that the present release of the *Valgarður* database will enhance the accessibility of the existing data and constitutes a valuable resource for future studies investigating the interplay between the physical and chemical evolution of rocks.

## 7 Sample availability

The detailed sample descriptions, combined with the photographs of many of the sample sites included as a supplement to the dataset, should facilitate future research visits to the investigated areas. Moreover, the remaining cores from the Orkustofnun samples are now kept at Náttúrufræðistofnun (the Icelandic Institute of Natural History) for interested parties to continue research.

## 8 Author contributions

Conceptualization: SWS, HF, LL, BG, JN, AV, MSG; Data curation: SWS, CC, LL, EJ, JF; Formal analysis: SWS, HF, LL, BG; Funding acquisition: MSG, SWS; Methodology: SWS, HF, LL, CC; Resources: EJ, BG, HF, MSG, JN, AV; Writing – original draft preparation: SWS, LL, BG, HF; Writing – review & editing: SS, LL, BG, HF, JF, MSG

## 9 Competing interests

The authors declare that they have no conflict of interest.

## 10 Acknowledgements

SWS and CC received funding from the Technical Development Fund of the Research Center of Iceland (RANNÍS, grant number 175 193-0612 Data Fusion for Geothermal Reservoir Characterization). This project additionally received funding

from Landsvirkjun Energy Research Fund (grant number NÝR-23-2019). We acknowledge Kristian Bär for sharing an early version of the P³ PetroPhysical Property Database (Bär et al., 2020). The database represents the collective efforts of many scientists over many decades – we would like to gratefully acknowledge the contribution of everyone who helped in sample collection, measurement, and data management. The manuscript was improved by the comments of two anonymous reviewers.

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
