# Peer review of "Valgarður: A Database of the Petrophysical, Mineralogical, and Chemical Properties of Icelandic Rocks"

_Earth System Science Data, 2022_

## Author Response (AR1)

We thank the reviewers for their detailed and helpful suggestions on the manuscript, which led to significant revisions in the manuscript. Most notably, both reviewers requested that in addition to describing the methods used to acquire the data, we present the data in the form of a Results section. We were initially hesitant during submission to include too much presentation and discussion of the data, as it significantly broadens the scope of the manuscript. However, in response to the comment from both reviewers, a Results and Discussion section has been added to the manuscript. As this section consists of 1 additional table, 6 additional figures, and 8 pages of text, it significantly lengthens the manuscript.

Below we address the reviews point-by-point, with the original comments posted in grey and the responses in red.

**Comments from Reviewer 1:**

The above-mentioned article describes the compilation of petrophysical, mineralogical and chemical properties of Icelandic rocks that have been analyzed over the past five decades. The paper describes the different measurement methods in greater detail. The database is presented in an EXCEL file and comprises five worksheets: 1) Petrophysical properties, 2) Mineralogical and geochemical data, 3) Photographs and sample sites, 4) Additional hyaloclastite data and 5) Extended references. Thereby, the photographs of the samples and sample locations are provided in an external zip file. In general, the article is well-written and clearly structured. However, several aspects should be considered prior publication in Earth System Science Data. Further comments below.

General comments regarding the manuscript:

The database presented here represents an updated version of an already published database available under Orkustofnun (2018; unfortunately, the provided link didn't work), which mainly contains porosity and permeability data analyzed on 529 samples. The dataset has been expanded in this manuscript by adding data from 302 samples collected by Orkustofnun between 1970-1980, and data from 161 samples retrieved from literature. Data analyzed on 34 samples are actually new and have not been published before.

The structure of the database is adapted from the P³ PetroPhysical Property database by Bär et al. (2020), which has been published here in ESSD. Some data has been even retrieved from this database; but not all data related to Iceland that is available in this repository were included. In the past two more rock property databases have been published here in ESSD:

Weinert et al. (2021): Database of petrophysical properties of the Mid-German Crystalline Rise, Earth Syst. Sci. Data, 13, 1441–1459, https://doi.org/10.5194/essd-13-1441-2021. - This database comprises almost 27000 single data points measured on plutonic rocks

Weydt et al. (2021): Petrophysical and mechanical rock property database of the Los Humeros and Acoculco geothermal fields (Mexico), Earth Syst. Sci. Data, 13, 571–598, https://doi.org/10.5194/essd-13-571-2021. - This database includes more than 31000 data points measured on volcanic, sedimentary, magmatic and metamorphic rocks

The input of the data is similar to Weydt et al. (2021), where in contrast to Bär et al. (2020) the data are stored in one row for each sample.

Two major point of criticism can be derived from this:

- What is actually new about this database? The authors should explain why it is necessary to have an extra database just for Icelandic rocks, also considering the impact factor of this

Journal and that the data included in this manuscript has already been published and discussed elsewhere.

While it is true that the layout and contents of the P³ PetroPhysical Property database (Bär et al., 2020) served as an inspiration for layout and contents of the database (in particular the structure of the Excel file), we would like to clarify that we did not "retrieve" any data in our database from Bär et al. (2020). The data in question is most of the samples from the report of Gudmundsson et al (1995) and Franzson et al. (2011). In our database, the data for these samples was retrieved from the original Valgarður database (an Access database – link provided in manuscript is correct), as well as the primary sources (reports written in Icelandic). In addition to petrophysical data for these samples, which is provided in Bär et al. (2020), the present database also contains detailed location and rock descriptions (provided in English and Icelandic), point counting, bulk-rock geochemical data, as well as acoustic velocities and mechanical properties (for measured samples), all which was missing from Bär et al. (2020).

With regards to the question of what is "new" about the database, we refer the reviewer to Table 1, as we added many additional sources of data other than Gudmundsson et al. (1995) and Franzson et al. (2011) to the database and made it easily accessible in one place. While it is true that much of this data has been previously published in the form of figures, this is the first time that this data has been rendered in this form, readily usable for scientists as well as the generic public. This database goes beyond the existing publications and conference papers that have cited this data but have mostly not provided it in tabulated form, let alone as a downloadable file.

We believe that the database serves as a useful stand-alone reference for Iceland, which is a country of tremendous geologic interest due to the active volcanism and widespread use of geothermal energy. Regarding the reviewer's comment questioning the need for an Iceland-specific database, they cite two previous databases published in ESSD that are highly localized -- one comprising two geothermal fields in Mexico and another for the mid-German Rise. We believe that ESSD is the proper venue for this database.

Bär, K., Reinsch, T., and Bott, J.: The PetroPhysical Property Database (P³) – a global compilation of lab-measured rock properties, Earth Syst. Sci. Data, 12, 2485–2515, https://doi.org/10.5194/essd-12-2485-2020, 2020.

- Although the authors mention the usage of the P³ database as a template in the acknowledgements, it is not mentioned in the manuscript. In my opinion it is necessary to put the development of this database in the right context and to correctly cite the source material as well as other important databases related to rock properties and geochemical data (e.g., Gard et al., 2019).

As discussed above, we do not believe that it is fair to say that we failed to cite the source material, as Bär et al. (2020) did not constitute a source of data. We cite all of the source material in Table 1, as well as in the main database.

We did in fact cite the P³ PetroPhysical Property database in the first paragraph of the introduction but incorrectly cited the year – we wrote Bär et al. (2019), and in the new version this has been correctly cited as Bär et al. (2020). Our sincere apologies for that mistake. In order to better contextualize this database, we added the following sentence to the introduction on lines 35-48:

*In recent years, a growing number of databases providing detailed petrophysical (Bär et al., 2020; Weinert et al. 2021; Weydt et al., 2021; Heap and Violay, 2021) and geochemical (Gard et al., 2019; Cole et al., 2022; Harðardóttir et al., 2022) data have been published.*

Bär, K., Reinsch, T., and Bott, J.: The PetroPhysical Property Database (P³) – a global compilation of lab-measured rock properties, Earth Syst. Sci. Data, 12, 2485–2515, https://doi.org/10.5194/essd-12-2485-2020, 2020.

Cole, T. L., Torres, M. A., and Kemeny, P. C.: The Hydrochemical Signature of Incongruent Weathering in Iceland, J. Geophys. Res. Earth Surf., 127, e2021JF006450, https://doi.org/https://doi.org/10.1029/2021JF006450, 2022.

Gard, M., Hasterok, D., and Halpin, J. A.: Global whole-rock geochemical database compilation, Earth Syst. Sci. Data, 11, 1553–1566, https://doi.org/10.5194/essd-11-1553-2019, 2019.

Harðardóttir, S., Matthews, S., Halldórsson, S. A., and Jackson, M. G.: Spatial distribution and geochemical characterization of Icelandic mantle end-members: Implications for plume geometry and melting processes, Chem. Geol., 120930, https://doi.org/ 10.1016/j.chemgeo.2022.120930, 2022.

Heap, M. J. and Violay, M. E. S.: The mechanical behaviour and failure modes of volcanic rocks: a review, Bull. Volcanol., 83, https://doi.org/10.1007/s00445-021-01447-2, 2021b.

Weinert, S., Bär, K., and Sass, I.: Database of petrophysical properties of the Mid-German Crystalline Rise, Earth Syst. Sci. Data, 13, 1441–1459, https://doi.org/10.5194/essd-13-1441-2021, 2021.

Weydt, L. M., Ramírez-Guzmán, Á. A., Pola, A., Lepillier, B., Kummerow, J., Mandrone, G., Comina, C., Deb, P., Norini, G., Gonzalez-Partida, E., Ramón Avellán, D., Maciás, J. L., Bär, K., and Sass, I.: Petrophysical and mechanical rock property database of the Los Humeros and Acoculco geothermal fields (Mexico), Earth Syst. Sci. Data, 13, 571–598, https://doi.org/10.5194/essd-13-571-2021, 2021.

Further points of criticism are:

- The lack of thermal properties and the small number of mechanical parameters. Especially the mechanical parameters would be important for models regarding volcanic activities/eruption/ flank stability as described in the introduction. Furthermore, Iceland is well-known for their geothermal fields. Thermal properties would be beneficial for, e.g., accurate temperature models.

More mechanical data was added to the database (Arngrímsson and Gunnarsson, 1999). In the new version of the manuscript, we describe how this data was collected in more detail, and present the mechanical data in Figures 14 and 15. With regards to thermal properties, we added thermal conductivity data for ~50 samples in the database, most of which derive from a single lava flow in the Reykjavik area (Guðlaugsson, 2000). We added further discussion of the point on lines 685-690:

> Thermal conductivity measurements are only available for a relatively small number of samples, most of which were derived from a single lava flow in the Reykjavik area (Guðlaugsson, 2000). Other studies have measured the thermal properties of Icelandic rocks (Ruether, 2011), and thermal conductivity and thermal diffusivity was measured on a large number of samples obtained from a nearly 2 km long core the Reyðarfjörður region (Oxburgh and Agrell, 1982; Drury, 1985; Flovenz and Saemundsson, 1993). However, to the authors best knowledge, the data obtained in these studies does not exist in tabulated form, at least accessible over the internet.

However, we would also like to note that thermal conductivity varies over a relatively narrow range (1.5-2.5 W m$^{-1}$ K$^{-1}$), which pales in comparison to the more than 1 order of magnitude variation in porosity and ~8 order of magnitude variation in permeability. Since heat transport in high-enthalpy volcanic geothermal fields is generally controlled by fluid advection rather than thermal conduction

(e.g., Ingebritsen et al., 2006), the thermal conductivity is of second-order importance in comparison to permeability when calculating the subsurface temperature distribution in high-enthalpy geothermal areas using numerical simulation packages such as TOUGH2. However, the data is more important when modeling conductive heat fluxes outside of volcanically-active areas (Flovenz and Saemundsson, 1993).

Arngrímsson, H. Ö. and Gunnarsson, Þ. B.: Tunneling in Acidic, Atered and Sedimentary Rock in Iceland - Búðarhálsvirkjun, Master's thesis, Technical University of Denmark, 162 pp., 2009.

Drury, M. J.: The Iceland research drilling project crustal section: physical properties of some basalts from the Reydarfjordur borehole, Iceland., Can. J. Earth Sci., 22, 1588–1593, https://doi.org/10.1139/e85-167, 1985.

Flóvenz, Ó. G. and Saemundsson, K.: Heat flow and geothermal processes in Iceland, 225, 123–138, https://doi.org/10.1016/0040-1951(93)90253-G, 1993.

Guðlaugsson, S. Þ.: An usual permeability anomaly in a Pleistocene shield-lava in Öskuhlið, Iceland - A study based on empirical relationships between petrophysical parameters, mineralogy and chemical composition, Orkustofnun, Reykjavik, Iceland, SÞG 01/00, 2000.

Ingebritsen, S. E., Sanford, W., and Neuzil, C. E.: Groundwater in Geologic Processes, Second Edi., Cambridge University Press, New York, 2006.

Oxburgh, E. R. and Agrell, S. O.: Thermal conductivity and temperature structure of the Reydardjordur borehole (Iceland)., J. Geophys. Res., 87, 6423–6428, https://doi.org/10.1029/JB087iB08p06423, 1982.

Ruether, J.: The Validation of the LMC device: Analysis of Icelandic basaltic rocks, Master's thesis, The School for Renewable Energy Science, 2011.

- The database represents a compilation of datasets obtained over five decades. Thus, different measurement devices and methods have been used on different sample sets. How can you ensure the comparability and reproducibility of the results? Would a data quality control system be useful?

In the paper, we describe in some detail the different measurement types and possible sources of bias related to different methodologies (e.g., Fig. 4). In the revised version of the text, we further developed these points and discussed sources of error with regards to connected porosity measurements on lines 250-264:

Connected porosities measured using gas expansion and triple weighing methods yield similar results, at least within a margin of uncertainty of <2-5%. This is demonstrated in Figure 3a, which reports connected porosity data collected by both methods on core samples from Krafla (Lévy et al., 2018, 2020b). For gas expansion measurements, an additional source of uncertainty is related to the choice of the saturating gas. Figure 3b compares connected porosity measurements performed on a set of hyaloclastite samples using either He (Franzson et al., 2011) or air (Frolova et al., 2005) as the saturating gas, and shows that connected porosity of samples measured using air is lower, likely due to the lesser ability of air to penetrate the microporosity or due to adsorption of water contained in the air into the clay-rich, altered rock. An additional source of error is that helium (or air) pycnometry requires the sample dimensions, whereas the triple-weight method only uses measurements of weight. As laboratory measurements of weight are often more accurate than measurements of length, this is one advantage of the triple-weight method over helium pycnometry. However, repeat

measurements of connected porosity on different core plugs obtained from a given rock outcrop (e.g. Fig. 1d) reveal natural uncertainty in the sampled rock of 5-10% (i.e. porosity can range from 5-15% for a given lava flow). Particularly for rocks such as hyaloclastites or flow-top breccias, which show strong gradients in petrophysical properties over distances <1 m, thus the uncertainty resulting from different measurement devices and methods Is likely less than the natural variability present in the rock.

We believe that a data quality control system would require subjective judgments about the quality of the data that might not be very well grounded. Instead, we provide detailed information the measurement types used to analyze each type of petrophysical in a column accompanying each data point. Users of the database can use this information to assess whether samples should be included or excluded in their analyses.

Franzson, H., Guðfinnsson, G. H., Frolova, J., Helgadóttir, H. M., Mortensen, A. K. and Jakobsson, S. P. Icelandic Hyaloclastite Tuffs. Iceland Geosurvey, ÍSOR-2011/064, 2011.

Frolova, J. V., Ladygin,, V. M., Franzson, H., Sigurðsson, O., Stefánsson, V. and Shustrov, V.: Petrophysical properties of fresh to mildly altered hyaloclastite tuffs, in: Proceedings World Geothermal Congress, Antalya, Turkey, 24-29 April 2005, 2005.

Lévy, L., Gibert, B., Sigmundsson, F., Flóvenz, O. G., Hersir, G. P., Briole, P., and Pezard, P. A.: The role of smectites in the electrical conductivity of active hydrothermal systems: Electrical properties of core samples from Krafla volcano, Iceland, Geophys. J. Int., 215, 1558–1582, https://doi.org/10.1093/gji/ggy342, 2018.

Lévy, L. E., Gibert, B., Escobedo, D., Patrier, P., Lanson, B., Beaufort, D., Loggia, D., Pezard, P. A., and Marino, N.: Relationships between lithology, permeability, clay mineralogy and electrical conductivity in Icelandic altered volcanic rocks, in: Proceedings World Geothermal Congress 2020+1, Reykjavik, Iceland, April-October, 2020, 2020b.

- Most of the properties have been analyzed on different sample sets. Thus, it is not possible to derive parameter relationships from this dataset, which in turn is crucial for reservoir parametrization (e.g., THCM modeling) and stochastic models as claimed in the conclusions. I agree that having some data is better than no data, but in the end this fact limits reservoir property estimations significantly. The authors should address this in a chapter regarding the limitations of this database and better explain the usage of the data.

We strongly disagree that "it is not possible to derive parameter relationships from this dataset". For example, Table 4 clarifies effective porosity and intrinsic permeability have been jointly analyzed on ~500 samples, nearly half of the database. The parametric relationship between porosity and permeability is of crucial importance for THMC modeling. To better clarify the ability of the database to illuminate the relationships between different petrophysical properties, lithologies and alteration zones, in the new version of the manuscript we have added a results/discussion section that presents the data in some detail.

- The paper includes a few figures that contain measurement results. Since the database does not contain too many datapoints (~5000 data points in total based on Table 4 and additional rock mechanical data), it could be beneficial to include a results section and to actually represent the data in greater detail. The authors could discuss the results as a kind of review. Thus, they would provide something new compared to the first version of the Valgardur database.

During initial preparation of the manuscript, we felt that full presentation of the data and discussion of the implications for understanding the relationships between lithology, alteration, and petrophysical properties was beyond the scope of the manuscript. As both reviewer 1 and reviewer 2 commented that it would be good to represent the detail in more detail, we have added a section to the manuscript that presents the petrophysical data in some detail. This significantly lengthens the manuscript. However, full description of the petrophysical, mineralogical and geochemical data is beyond the scope of this manuscript.

- The results of the VS measurements in Fig. 8 show a high scatter. The authors should check this data for outliers and maybe plot also VP vs VS.

We believe that the scatter in the Vp and Vs measurements is real, related to variability in the nature of porosity (e.g. pore vs. microcrack-dominated) and hydrothermal alteration. In the new version of the manuscript, we provided more discussion of sources of variability in these measurements on lines 650-663:

[revised manuscript text omitted]

- Please improve Fig 1, particularly Fig. 1e.

We have increased the size of Figure 1 and fixed Fig. 1e.

- Please be careful with the references. Some articles are cited in the manuscript, but are not included in the reference list and vice versa. Also, the formatting is inconsistent and should be corrected according to the journal's standards.

We have gone carefully over the references, and have tried our best to maintain consistency with the ESSD reference formatting.

Comments regarding the database:

Besides some metadata (references, sample location and descriptions), the first worksheet "Petrophysical properties" contains the results of the different petrophysical measurements. In general, the table is well-structured and the columns are highlighted in different colours to increase the readability. I appreciate the detailed lithological information and the possibility to distinguish between unaltered and altered rocks and to be able to compare this information with the photographs. However, the quality of the sample descriptions varies a lot between the different datasets. Furthermore, the sample IDs are sometimes just numbers (varying from 4 to 150803-03) and in other cases a combination of letters and numbers. It would be useful to use a homogenized sample ID for the database and eventually keep the original ID in a separate column as it has been done in the P³ database in Bär et al. (2020).

We adapted the suggestion of the reviewer and added an additional column with a homogenized Sample ID, with the identification of the study as well as the original sample number. We note in the text on lines 154 that the different studies provide different levels of detail in the sample descriptions. In the case of many borehole samples, sample descriptions were obtained from well reports.

The second worksheet "Mineralogical and geochemical data" comprises a comprehensive dataset of major and trace elements, quantitative XRD data and point counting data retrieved from thin section analyses. Likewise, the columns are highlighted in different colours to increase the readability. I appreciate the detailed thin section analysis, which is often missing in other petrophysical datasets. I would suggest to include the rock types here again to make it easier for the user (you don't need to switch between the sheets). Unfortunately, RRE data is not available here.

We added the suggestion of the reviewer and added the basic rock types and alteration zones to this worksheet. Note that there is trace element data provided, including several rare earth elements (e.g. Y, Ce, Sc).

The third sheet only contains the numbers of the photographs and the associated sample IDs. About 140 photos are provided in the additional zip folder. However, as a stand-alone worksheet this looks less impressive. I understand that this database is build according to the P³ database and shall resemble a relational database, but additional information such as lithology and outcrop name would improve this worksheet.

We adapted the suggestion of the reviewer and added the basic rock types and alteration zones, as sample description, to this worksheet. However, we moved this worksheet out of the main file and added it to a separate file.

The fourth worksheet includes "Additional hyaloclastite data" and represents a short version of worksheet 1. This table includes a smaller list of rock properties and it is not clear to me why this data is not included in the main data sheet. The different measurement methods as described in line 102 on page 6 could be easily marked in worksheet 1. Furthermore, the list should be edited and be consistent with worksheet 1.

The main reason this data was included was because measurements of connected porosity using air or He were compared in Fig. 3b. While only the He-porosimetry measurements are reported in the

database, as suggested by the reviewer, all the rest of the data has been included in the main data sheet. Therefore, this tab was removed from the database.

The last worksheet includes the references of the original data sources. However, this could be listed in a PDF file and/or in a much prettier way.

We adapted the suggestion of the reviewer and removed the list of references into a pdf file uploaded with the most recent version of the repository.

While there is a "READ ME" sheet at the beginning of the database that contains a brief description of each worksheet, headings and additional information about each table in the various worksheets would improve navigation and orientation in the Excel file.

We have provided much more detail regarding the contents of each worksheet in the "READ ME" sheet. All necessary information regarding column headings and units is provided in the "READ ME" sheet as well as in each of the main work sheets.

**Comments from Reviewer 2:**

The submitted manuscript presents a database—called Valgarður—that contains petrophysical, mineralogical, and chemical properties of rocks from Iceland. Although there are gaps in the database, such as the lack of mechanical and thermal properties, I think that it will prove to be a useful resource. I recommend publication after the following comments have been addressed to the satisfaction of the editor.

Line 70: Valgarður.

Fixed

Figure 2: "Kr" is Krafla?

Added

Table 2: I think it's odd to describe lavas as "fine-medium grained" and "medium-coarse grained". Lavas do not contain grains, but crystals. Would aphanitic and porphyritic be more appropriate descriptors?

We already used the descriptor porphyritic for porphyritic basaltic lavas. Although the reviewer's point is well received, it has been customary in Iceland to describe basalts in terms of "grain" size. We added more discussion around this point to the text on lines 164-167:

> It has been customary in Iceland to classify lava flows based on crystal size (Guðmundsson et al., 1995); in comparison to the classification scheme of Walker (1958), the lithological identifier "fine-medium grained basaltic lava" generally corresponds to the "tholeiitic basalt" type, while "medium-coarse grained basaltic lava" corresponds to the "olivine basalt" type.

Walker, G. P. L.: Geology of the Reydarfjördur area, Eastern Iceland, Quarterly Journal of the Geological Society of London, 114, 367–391, https://doi.org/10.1144/gsjgs.114.1.0367, 1958.

Line 153: The authors could refer here to the lava shown in Figure 1.

We added the reference to Fig. 1c for the porphyritic basaltic lava flow.

Line 165: Suggest to change "ash flows" to "pyroclastic density currents".

Suggestion adopted.

Line 175: Can the authors refer the reader to the paper(s) describing these rocks, or are these observations unique to this study?

We added citations describing these kinds of rocks as well as further clarification regarding volcaniclastic rocks and sedimentary rocks on lines 189-193:

> While many volcaniclastic rocks (e.g., hyaloclastite) show sedimentary textures related to deposition in a sub-aqueous or sub-aerial environment (Bergh and Sigvaldason, 1991; Schopka et al., 2006; Banik et al., 2014; Greenfield et al., 2020), they are not referred to as sedimentary rocks in the database. Examples of sedimentary rocks found in Iceland include clay-rich lacustrine sediments, glacial tillite and conglomerates, sandstone, as well as interbasaltic beds (e.g. Bennet et al., 2000; Arnalds et al., 2001; Thorpe et al., 2019; Eiriksson and Simonarson, 2021).

Arnalds, O., Gisladottir, F. O., and Sigurjonsson, H.: Sandy deserts of Iceland: an overview, J. Arid Environ., 47, 359–371, https://doi.org/https://doi.org/10.1006/jare.2000.0680, 2001.

Bergh, S. G. and Sigvaldason, G. E.: Pleistocene mass-flow deposits of basaltic hyaloclastite on a shallow submarine shelf, South Iceland, Bull. Volcanol., 53, 597–611, https://doi.org/10.1007/BF00493688, 1991.

Banik, T. J., Wallace, P. J., Höskuldsson, Á., Miller, C. F., Bacon, C. R., and Furbish, D. J.: Magma–ice–sediment interactions and the origin of lava/hyaloclastite sequences in the Síða formation, South Iceland, Bull. Volcanol., 76, 785, https://doi.org/10.1007/s00445-013-0785-3, 2013.

Bennett, M. R., Huddart, D., and McCormick, T.: An integrated approach to the study of glaciolacustrine landforms and sediments: a case study from Hagavatn, Iceland, Quat. Sci. Rev., 19, 633–665, https://doi.org/10.1016/S0277-3791(99)00013-X, 2000.

Greenfield, L., Millett, J. M., Howell, J., Jerram, D. A., Watton, T., Healy, D., Hole, M. J., and Planke, S.: The 3D facies architecture and petrophysical properties of hyaloclastite delta deposits: An integrated photogrammetry and petrophysical study from southern Iceland, Basin Res., 32, 1091–1114, https://doi.org/10.1111/bre.12415, 2020.

Schopka, H. H., Gudmundsson, M. T., and Tuffen, H.: The formation of Helgafell, southwest Iceland, a monogenetic subglacial hyaloclastite ridge: Sedimentology, hydrology and volcano–ice interaction, J. Volcanol. Geotherm. Res., 152, 359–377, https://doi.org/https://doi.org/10.1016/j.jvolgeores.2005.11.010, 2006.

Thorpe, M. T., Hurowitz, J. A., and Dehouck, E.: Sediment geochemistry and mineralogy from a glacial terrain river system in southwest Iceland, Geochim. Cosmochim. Acta, 263, 140–166, https://doi.org/https://doi.org/10.1016/j.gca.2019.08.003, 2019.

Line 194: Presumably the measurements of permeability were made under a small confining pressure?

This is true and clarified in lines 218-219:

> Permeability measurements were made under a variable but low confining pressure (<5 MPa).

Table 4: Acronyms should be explained in the table caption.

Acronyms have been added to the table caption.

Line 203: I suggest to add that "effective porosity" is often called "connected porosity". This porosity is that connected to the outside surface of the sample. I suggest that the authors refine their definition here.

We have adapted the suggestion of the reviewer and changed "effective" porosity to "connected porosity" throughout the database and the manuscript. We refined our definition of connected porosity on lines 230-231: "Porosity is differentiated between connected porosity (the fraction of bulk volume occupied by pore space connected to the outside surface of the sample; this is also referred to as "effective porosity)…"

Line 218: Suggest to change "brine" to "liquid".

Suggestion adapted.

Line 224: It might be worth noting that this may not be the case for clay-rich, altered materials.

Suggestion adapted.

Line 225: Another source of error is that helium pycnometry requires the sample dimensions. The triple-weight method, however, only uses measurements of weight. Laboratory measurements of weight are often more accurate than measurements of length, which is another advantage of the triple-weight method over helium pycnometry.

We have added additional discussion of the uncertainty related to pynometry, triple-weighing, and compared this with the natural uncertainty present in the rock on lines 253-264:

Figure 3b compares connected porosity measurements performed on a set of hyaloclastite samples using either He (Franzson et al., 2011) or air (Frolova et al., 2005) as the saturating gas, and shows that connected porosity of samples measured using air is lower at low porosities and higher at high porosities. While the former may be due to the lesser ability of air to penetrate the microporosity, the latter may result from adsorption of water contained in the air in the clay-rich, altered rock. An additional source of error is that helium (or air) pycnometry requires the sample dimensions, whereas the triple-weight method only uses measurements of weight. As laboratory measurements of weight are often more accurate than measurements of length, this is one advantage of the triple-weight method over helium pycnometry. However, repeat measurements of connected porosity on different core plugs obtained from a given rock outcrop (e.g. Fig. 1d) reveal natural uncertainty in the sampled rock of 5-10% (i.e. porosity can range from 5-15% for a given lava flow). Given the natural variability and heterogeneity present in the rock, particularly hyaloclastites or flow-top breccias, which show strong gradients in petrophysical properties over distances <1 m, the uncertainty resulting from different measurement devices and methods is likely less than (or comparable to) the natural variability present in the rock.

Franzson, H., Guðfinnsson, G. H., Frolova, J., Helgadóttir, H. M., Mortensen, A. K. and Jakobsson, S. P. Icelandic Hyaloclastite Tuffs. Iceland Geosurvey, ÍSOR-2011/064, 2011.

Frolova, J. V., Ladygin,, V. M., Franzson, H., Sigurðsson, O., Stefánsson, V. and Shustrov, V.: Petrophysical properties of fresh to mildly altered hyaloclastite tuffs, in: Proceedings World Geothermal Congress, Antalya, Turkey, 24-29 April 2005, 2005.

Line 227: Unless dried/filtered, air can also contain water, which can also influence measurements of porosity (especially clay-rock, altered materials).

This is mentioned in the text; see response to comment regarding line 225.

Line 229: Given the variability and heterogeneity of volcanic rock, this seems very likely.

This is mentioned in the text; see response to comment regarding line 225.

Line 250: Presumably this is because the alteration mineral assemblage formed at depth contains denser minerals? Is this true?

The alteration mineralogy assemblage between these sets of samples is similar (both are smectite-zeolite zone), but the alteration mineral assemblage may be denser in response to compaction and increasing confining stress. This is now on lines 277-283:

> Figure 4 shows that the range of grain density measured in smectite-zeolite altered lava flows is similar whether Hg displacement (blue lines) or triple-weighting (red lines) techniques are used (Fig. 4a). On the other hand, for altered hyaloclastites (Fig. 4b), samples analyzed by triple-weighting ($\sim$2.75 g cm$^{-3}$) show significantly larger average grain density than by Hg displacement ($\sim$2.6 g cm$^{-3}$). However, other factors might also explain this discrepancy, most notably that many of the samples shown in Fig. 4b analyzed using Hg displacement were derived from surface outcrops, whereas those analyzed by triple-weighting were obtained from borehole samples in active geothermal systems. Although these samples are in similar alteration zones, rocks at depth are more compacted, and may contain a denser alteration mineral assemblage.

Line 259: Higher, and also much more accurate. I think that the authors should clearly state that estimating porosity using thin section images is problematic and often underestimates the porosity. For example, it is often not possible to identify micropores and small microcracks on thin section images, which can form a large proportion of the porosity in some volcanic rocks.

We have taken the reviewers suggestion, and made to revisions to the section concerning petrographic estimation of porosity and primary porosity on lines 282-297 and in Figure 5:

> Porosity was also assessed in 352 samples by point-counting. In point-counting, a thin section (around 30 micrometers thick) of the sample was prepared, and a regular grid with a given number of points (usually 200 or 1000 points) was arrayed onto the thin section image. Identification of mineralogy or pore space at each point was performed, with the different studies applying different levels of classification between primary minerals, glass, pore space, and alteration minerals. The primary porosity formed during magma emplacement and cooling was estimated as the sum of remaining open-space porosity in the rock as well as that of secondary alteration minerals that have precipitated into vesicles (Petford, 2003). This technique does not measure the cross-sectional area of microcracks, but rather only identifies macroscopic pores on the order of 1 mm or larger (Neuhoff et al., 1999; Manning and Bird, 1995; Chayes, 1956). Although the contribution of the latter to the porosity is often large, note that distinction between open-space porosity created by post-eruptive processes (fracturing/veining) was not made by all studies. Therefore, we estimate that the uncertainty of porosity measurements by point counting is considerable (>5-10%). Figure 5 shows that the remaining open-space porosity measured by point counting is generally lower than that obtained by gas expansion, particularly for altered, high-porosity hyaloclastites, indicating the dominance of microporosity on the total porosity. However, a significant number of samples ($\sim$50) had higher porosity recorded using petrographic analysis than by gas expansion. As porosity and permeability are scale-dependent (Manning and Bird, 1995), such variability

could also indicate natural heterogeneity in the rock and preparation of the thin section from a particularly compact or porous part of the material.

[Figure]

Figure 5. Remaining porosity estimated by point counting compared with connected porosity measured using He pycnometry.

Chayes, F.: Petrographic model analysis: an elementary statistical appraisal, John Wiley and Sons, New York, New York, United States of America, 1956.

Manning, C. E. and Bird, D. K.: Porosity, permeability and basalt metamorphism, 123–140, in: Low-Grade Metamorphism of Mafic Rocks, edited by: Schiffman, P. and Day, H.W., Geological Society of America Special Paper 296, https://doi.org/10.1130/SPE296-p123, 1995.

Neuhoff, P., Fridriksson, T., Arnórsson, S., and Bird, D. K.: Porosity evolution and mineral paragenesis during low-grade metamorphism of basaltic lavas at Teigarhorn, eastern Iceland, Am. J. Sci., 299, 467–501, 1999.

Petford, N.: Controls on primary porosity and permeability development in igneous rocks, Geol. Soc. London, Spec. Publ., 214, 93 LP – 107, https://doi.org/10.1144/GSL.SP.2003.214.01.06, 2003.

Line 264: What was the confining pressure used by Gudmundsson et al. (1995)?

Although the confining pressure used by Guðmundsson et al. (1995) is not clearly stated in the report, we assume that the confining pressure is relatively low based on the technical specifications of the CMS-300 device. We clarified this in the text on lines 302-305 and also added a reference where measurements at variable confining pressure were performed on a few samples:

> While permeability of the samples from Guðmundsson et al. (1995) were measured using a low (<4 MPa) confining pressure, measurements performed on a few samples under varying confining pressure showed little dependence of permeability on confining pressure (Johnson and Boitnott, 1998).

Johnson, J. & Boitnott, G. N.: Velocity, Permeability, Resistivity and Pore Structure Models of Selected Basalts from Iceland. New England Research, Vermont, U.S.A, 1998.

Line 265: What is meant by the "stationary method"? The steady-state flow method?

This has been changed to the steady-state flow method.

Line 287: The authors should also discuss the Forchheimer correction. This correction is often needed when measuring the permeability of porous rocks using gas. The data were also checked for the Forchheimer correction? If yes, the authors should discuss this here. If not, I think that the authors should clearly state that these data were not checked for the Forchheimer correction, and so may not represent the "true" or "intrinsic" permeability.

The data were checked for Forchheimer correction. More discussion of the Forchheimer correction has been added to the text on lines 329-338:

> However, in high permeability rocks, turbulent flow regimes may develop, and Darcy's law needs to be modified in order to take into additional flow resistance resulting from inertial forces, as given by the second term in the Forchheimer equation (Forchheimer, 1901; Zeng and Grigg, 2006):
>
> $$\frac{dp}{dx} = \frac{\mu}{k}\frac{q}{A} + \beta\left(\frac{q}{A}\right)^2 \tag{7}$$
>
> For the samples analyzed by Guðmundsson et al. (1995), the inertial coefficient $\beta$ was calculated for a sample using repeat measurements at different flow rates. For the samples analyzed by Levy et al. (2018, 2020b), the Forchheimer correction was not applied, and therefore the reported permeability may not be the actual rock permeability; however, as many of these samples are low permeability ($<10^{-15}$ m$^2$), turbulent conditions are unlikely to develop and the Forchheimer correction is likely not significant (Heap et al., 2018).

Forchheimer, P.: Wasserbewegung durch boden, Z. Ver. Deutsch, Ing., 45, 1782–1788, 1901.

Heap, M. J., Reuschlé, T., Farquharson, J. I., and Baud, P.: Permeability of volcanic rocks to gas and water, J. Volcanol. Geotherm. Res., 354, 29–38, https://doi.org/10.1016/j.jvolgeores.2018.02.002, 2018.

Lévy, L., Gibert, B., Sigmundsson, F., Flóvenz, O. G., Hersir, G. P., Briole, P., and Pezard, P. A.: The role of smectites in the electrical conductivity of active hydrothermal systems: Electrical properties of core samples from Krafla volcano, Iceland, Geophys. J. Int., 215, 1558–1582, https://doi.org/10.1093/gji/ggy342, 2018.

Lévy, L. E., Gibert, B., Escobedo, D., Patrier, P., Lanson, B., Beaufort, D., Loggia, D., Pezard, P. A., and Marino, N.: Relationships between lithology, permeability, clay mineralogy and electrical conductivity in Icelandic altered volcanic rocks, in: Proceedings World Geothermal Congress 2020+1, Reykjavik, Iceland, April-October 2020, 2020b.

Zeng, Z. and Grigg, R.: A Criterion for Non-Darcy Flow in Porous Media, Transp. Porous Media, 63, 57–69, https://doi.org/10.1007/s11242-005-2720-3, 2006.

Line 289: The authors should state/discuss whether these data are influenced by rock type. It's also interesting to note that the lava samples cover almost the entire permeability range.

Extensive discussion of the influence of rock type and alteration on permeability has been added to the results & discussion section in the new manuscript.

Line 289: What was the concentration of brine used?

We believe it was misleading to label these as "Brine permeabilities" as the salinity of the liquid was very low. We clarify this in the new version of the data base by using the term "Liquid apparent permeability" rather than "Brine apparent permeability" and moreover state clearly that low-salinity water was used for permeability testing.

Line 290: The authors should offer a reason for this here, in my opinion. This difference is often attributed to the presence of swelling clays (see, for example, Tanikawa and Shimamoto, 2009). Even in clay-free volcanic rocks, liquid permeabilities can also be lower than gas permeabilities due to water adsorption on narrow, tortuous microstructural elements (see Heap et al., 2018).

Tanikawa, W., & Shimamoto, T. (2009). Comparison of Klinkenberg-corrected gas permeability and water permeability in sedimentary rocks. International Journal of Rock Mechanics and Mining Sciences, 46(2), 229-238.

Heap, M. J., Reuschlé, T., Farquharson, J. I., & Baud, P. (2018). Permeability of volcanic rocks to gas and water. Journal of Volcanology and Geothermal Research, 354, 29-38.

Further discussion of this point has been added to the text on lines 339-346:

> Figure 6a shows that apparent permeability measured using air generally exceeds that measured using water, often by several orders of magnitude.  This difference is often attributed to the presence of swelling clays (see, for example, Tanikawa and Shimamoto, 2009). Even in clay-free volcanic rocks, liquid permeabilities can also be lower than gas permeabilities due to water adsorption on narrow, tortuous microstructural elements (Heap et al., 2018). Measurements of air permeability that are less than brine permeability are generally considered unreliable. Figure 6b compares air apparent permeability and intrinsic permeability, showing that the magnitude of the Klinkenberg correction increases with decreasing permeability, consistent with increased gas slippage during flow through microstructural elements in low-porosity, low-permeability rock (Heap et al., 2018).

Line 292: Is it not worth adding a plot that shows permeability as a function of porosity? It would be interesting to show whether permeability increases as a function of porosity, as seen in, for example, Farquharson et al. (2015). Is it worth adding another plot that differentiates the data by their alteration?

Farquharson, J., Heap, M. J., Varley, N. R., Baud, P., & Reuschlé, T. (2015). Permeability and porosity relationships of edifice-forming andesites: a combined field and laboratory study. Journal of Volcanology and Geothermal Research, 297, 52-68.

A plot showing the relationship of porosity and permeability to lithology and alteration have been added to the results section (Fig. 11).

[Figure]

**Figure 11. Relationship between connected porosity and intrinsic permeability in a) lava flows, b) hyaloclastites and pillow basalts, and c) other lithologies, including basaltic intrusions, silicic intrusions, silicic volcanics, intermediate rocks, and sediments. Samples colored by alteration zone, with symbols corresponding to different lithology.**

Line 353: Why not show formation factor and/or surface conductivity as a function of porosity for the data in the database?

Plots showing the formation factor as a function of porosity have been added to the results section (Figure 12).

[Figure]

**Figure 12. a) Relationship between grain density and bulk resistivity. Smectite-rich rocks which usually constitute the cap rock have a lower resistivity and grain density than rocks that compose the resistive core or fresh, unaltered rocks. b) Relationship between formation resistivity factor and connected porosity. Note that the outliers used the apparent formation resistivity factor, i.e. calculated using only a single salinity (see text).**

Lines 356-357: These acronyms have already been defined above.

We removed the definitions of these acronyms from this part of the text.

Line 370: Is there a reference for these standard techniques?

We provided more detail concerning used XRF techniques in the new version of the manuscript on lines 410-414:

> Bulk rock chemical analyses were performed by two commercial chemical laboratories, The Caleb Brett Laboratory in England and McGill University in Canada, using standard XRF techniques (e.g., Potts and Webb, 1992; Rousseau et al., 1996). Both labs used the fused bead technique for major elements and pressed powder pellets for the determination of trace elements. Values for samples analyzed by both laboratories are generally within analytical error (Rousseau et al., 1996).

Potts, P. J. and Webb, P. C.: X-ray fluorescence spectrometry, J. Geochemical Explor., 44, 251–296, https://doi.org/https://doi.org/10.1016/0375-6742(92)90052-A, 1992.

Rousseau, R. M., Willis, J. P., and Duncan, A. R.: Practical XRF Calibration Procedures for Major and Trace Elements, 25, 179–189, https://doi.org/10.1002/(SICI)1097-4539(199607)25:4<179::AID-XRS162>3.0.CO;2-Y, 1996.

Line 410: Is it worth adding another plot that differentiates the data by their alteration? Or providing a plot that shows that the saturated velocities are higher than the dry velocities?

We added a plot that differentiates the data by alteration and also compares saturated and dry velocities in the new version of the manuscript (Fig. 13).

[Figure]

**Figure 13.** **a. Compressional (P-wave) velocities under dry (transparent symbols) and saturated (opaque symbols) conditions versus connected porosity. b. S-wave velocities versus P-wave velocities under saturated conditions.**

Line 411: I think it would help to state that this trend of often seen for rocks, including volcanic rocks (with references).

This suggestion was adapted in the new results section, on lines 660-654:

> Figure 13a shows that P-wave velocities are typically inversely correlated to porosity: crystalline basalts show the highest velocities and the lowest porosities, while hyaloclastites have the lowest velocities and higher porosities. This relationship has been seen in several previous studies of volcanic rocks (e.g., Pola et al., 2014; Frolova et al., 2014; Wyering et al., 2014; Heap et al., 2015; Durán et al., 2019; Frolova et al., 2021).

Durán, E. L., Adam, L., Wallis, I. C., and Barnhoorn, A.: Mineral Alteration and Fracture Influence on the Elastic Properties of Volcaniclastic Rocks, J. Geophys. Res. Solid Earth, 124, 4576–4600, https://doi.org/10.1029/2018JB016617, 2019.

Frolova, J., Ladygin, V., Rychagov, S., Zukhubaya, D. Effects of hydrothermal alterations on physical and mechanical properties of rocks in the Kuril-Kamchatka island arc, Eng. Geol. 183, 80–95, https://doi.org/10.1016/j.enggeo.2014.10.011, 2014.

Frolova, J. V, Chernov, M. S., Rychagov, S. N., Ladygin, V. M., Sokolov, V. N., and Kuznetsov, R. A.: The influence of hydrothermal argillization on the physical and mechanical properties of tuffaceous rocks: a case study from the Upper Pauzhetsky thermal field, Kamchatka, Bull. Eng. Geol. Environ., 80, 1635–1651, https://doi.org/10.1007/s10064-020-02007-2, 2021.

Heap, M. J., Kennedy, B. M., Pernin, N., Jacquemard, L., Baud, P., Farquharson, J. I., Scheu, B., Lavallée, Y., Gilg, H. A., Letham-Brake, M., Mayer, K., Jolly, A. D., Reuschlé, T., and Dingwell, D. B.: Mechanical behaviour and failure modes in the Whakaari (White Island volcano) hydrothermal system, New Zealand, J. Volcanol. Geotherm. Res., 295, 26–42, https://doi.org/https://doi.org/10.1016/j.jvolgeores.2015.02.012, 2015.

Pola, A., Crosta, G. B., Fusi, N., and Castellanza, R.: General characterization of the mechanical behaviour of different volcanic rocks with respect to alteration, Eng. Geol., 169, 1–13, https://doi.org/10.1016/j.enggeo.2013.11.011, 2014.

Wyering, L. D., Villeneuve, M. C., Wallis, I. C., Siratovich, P. A., Kennedy, B. M., Gravley, D. M., and Cant, J. L.: Mechanical and physical properties of hydrothermally altered rocks, Taupo Volcanic Zone, New Zealand, J. Volcanol. Geotherm. Res., 288, 76–93, https://doi.org/10.1016/j.jvolgeores.2014.10.008, 2014.

Lines 412-413: The authors should provide a reference in support of this statement. The scatter in these data is a result of the fact that porosity is just a scalar, and elastic wave velocities are sensitive to the nature of the porosity (microcracks versus pores).

We provide more detail as well as references on lines 660-663:

> As P- and S-wave velocities are strongly dependent on crack density and geometry, highly cracked rocks may display in some cases very low velocities at room conditions (e.g., Nur and Simmons, 1969; Vinciguerra et al., 2005; Nara et al., 2011). This could explain the low P-wave velocities seen in lava flows and hyaloclastites in Figure 13.

Guéguen, Y. and Palciauskas, V.: Introduction to the Physics of Rocks, Princeton University Press, 1994.

Nara, Y., Meredith, P. G., Yoneda, T., and Kaneko, K.: Influence of macro-fractures and micro-fractures on permeability and elastic wave velocities in basalt at elevated pressure, 503, 52–59, https://doi.org/10.1016/j.tecto.2010.09.027, 2011.

Nur, A. and Simmons, G.: The effect of saturation on velocity in low porosity rocks, Earth Planet. Sci. Lett., 7, 183–193, https://doi.org/10.1016/0012-821X(69)90035-1, 1969.

Wyering, L. D., Villeneuve, M. C., Wallis, I. C., Siratovich, P. A., Kennedy, B. M., Gravley, D. M., and Cant, J. L.: Mechanical and physical properties of hydrothermally altered rocks, Taupo Volcanic Zone, New Zealand, J. Volcanol. Geotherm. Res., 288, 76–93, https://doi.org/10.1016/j.jvolgeores.2014.10.008, 2014.

Vinciguerra, S., Trovato, C., Meredith, P. G., and Benson, P. M.: Relating seismic velocities, thermal cracking and permeability in Mt. Etna and Iceland basalts, Int. J. Rock Mech. Min. Sci., 42, 900–910, https://doi.org/10.1016/j.ijrmms.2005.05.022, 2005.

Lines 418-420: Although the data are few, I think the authors should offer more details as to how these data were collected and, briefly, describe the data obtained.

In the new version of the manuscript, we describe in more detail how the mechanical data was collected by the different studies on lines 489-497:

> For the hyaloclastites samples analyzed by Frolova et al. (2005) and Franzson et al. (2011), the uniaxial compressive strength test was performed by standard testing procedures in accordance with State Standards 21153.2-84 (1984) and ASTM D7012 (American Society for Testing Materials, 2013). Uniaxial compressive strength was measured using a German hydraulic press CDM-10/91 and was determined for samples in dry and water-saturated states. The samples analyzed by Árngrimsson and Gunnarsson (1999) were analyzed at the Technical University of Denmark (DTU), which performed triaxial tests on five samples, and the Danish Geotechnical Institute (GEO), which performed Brazil tests on 55 samples and unconfined compressive strength tests on 36 samples, with methods according to IRSM standard (Ulusay and Hudson, 2007). The elastic constants given for the samples from Frolova et al. (2005), Franzson and Tulinius (1999) and Jaya et al. (2010) were calculated from the measured wave velocities and the bulk density (e.g. Mavko et al., 2009).

American Society for Testing Materials. ASTM D7012-13. Standard test methods for compressive strength and elastic moduli of intact rock core specimens under varying states of stress and temperatures. American Society for Testing Materials, Pennsylvania, USA, 2013

Arngrímsson, H. Ö. and Gunnarsson, Þ. B.: Tunneling in Acidic, Atered and Sedimentary Rock in Iceland - Búðarhálsvirkjun, Master's thesis, Technical University of Denmark, 162 pp., 2009.

Franzson, H., and Tulinius, H.: Rannsóknir á kjarna úr holu ÖJ-1, Ölkelduhálsi (Research on core from hole ÖJ-1, Ölkelduháls), Orkustofnun (OS-99024), Reykjavik, Iceland, 1999.

Franzson, H., Guðfinnsson, G. H., Frolova, J., Helgadóttir, H. M., Mortensen, A. K. and Jakobsson, S. P. Icelandic Hyaloclastite Tuffs. Iceland Geosurvey, ÍSOR-2011/064, 2011.

Frolova, J. V., Ladygin, V. M., Franzson, H., Sigurðsson, O., Stefánsson, V. and Shustrov, V.: Petrophysical properties of fresh to mildly altered hyaloclastite tuffs, in: Proceedings World Geothermal Congress, Antalya, Turkey, 24-29 April 2005, 2005.

Jaya, M. S., Shapiro, S. A., Kristinsdóttir, L. H., Bruhn, D., Milsch, H., and Spangenberg, E.: Temperature dependence of seismic properties in geothermal rocks at reservoir conditions, Geothermics, 39, 115–123, https://doi.org/ 10.1016/j.geothermics.2009.12.002, 2010.

Mavko, G., Mukerji, T., and Dvorkin, J.: The Rock Physics Handbook: Tools for Seismic Analysis of Porous Media, 2nd ed., Cambridge University Press, Cambridge, https://doi.org/10.1017/CBO9780511626753, 2009.

State Standard 21153.2-84, 1984 bb. Rocks. Methods for determination of uniaxial compressive strength. Publisher of Standards, Moscow (12 pp.).

Ulusay R, Hudson JA (eds): The complete ISRM suggested methods for rock characterization, testing and monitoring: 1974–2006. International Society for Rock Mechanics Turkish National Group, Ankara, 2007

Line 443: I think it would also be interesting to measure thermal properties.

We made sure to add available data for thermal conductivity to the database (Franzson and Tulinius, 1999; Árngrimsson and Gunnarsson, 1999). In addition, in the new version of the manuscript we discuss thermal properties in more detail in the conclusions on lines 717-722:

> Thermal conductivity measurements are only available for a relatively small number of samples, most of which were derived from a single lava flow in the Reykjavik area (Guðlaugsson, 2000). Other studies have measured the thermal properties of Icelandic rocks (Ruether, 2011), and thermal conductivity and thermal diffusivity was measured on a large number of samples obtained from a nearly 2 km long core in the Reyðarfjörður region (Oxburgh and Agrell, 1982; Drury, 1985; Flovenz and Saemundsson, 1985). However, to the authors best knowledge, the data obtained in these studies does not exist in tabulated form, at least accessible over the internet.

Franzson, H., and Tulinius, H.: Rannsóknir á kjarna úr holu ÖJ-1, Ölkelduhálsi (Research on core from hole ÖJ-1, Ölkelduháls), Orkustofnun (OS-99024), Reykjavik, Iceland, 1999.

Drury, M. J.: The Iceland research drilling project crustal section: physical properties of some basalts from the Reydarfjordur borehole, Iceland., Can. J. Earth Sci., 22, 1588–1593, https://doi.org/10.1139/e85-167, 1985.

Flóvenz, Ó. G. and Saemundsson, K.: Heat flow and geothermal processes in Iceland, 225, 123–138, https://doi.org/10.1016/0040-1951(93)90253-G, 1993.

Ingebritsen, S. E., Sanford, W., and Neuzil, C. E.: Groundwater in Geologic Processes, Second Edi., Cambridge University Press, New York, 2006.

Oxburgh, E. R. and Agrell, S. O.: Thermal conductivity and temperature structure of the Reydardjordur borehole (Iceland)., J. Geophys. Res., 87, 6423–6428, https://doi.org/10.1029/JB087iB08p06423, 1982.

Ruether, J.: The Validation of the LMC device: Analysis of Icelandic basaltic rocks, Master's thesis, The School for Renewable Energy Science, 2011.

Line 446: "these important data"

Removed from new version of manuscript.

---

## Author Response (AR2)

In general, the authors have improved their work compared to the previous version published as a preprint in ESSD. The structure of the database is now much clearer and easier to understand. Here I noticed only a few editing errors that the authors may want to fix before the final publication (frame of Excel cells or two columns for thermal conductivity, but only one is backed with data). Most of the reviewers' comments have also been implemented. In particular, the authors have expanded the results section and included several diagrams with petrophysical data.

We thank the reviewer for their comments and suggestions. The extra column for thermal conductivity has been deleted from the Excel file.

There are only a few minor points that should be addressed prior to publication:

• Both reviewers mentioned the small number of mechanical data and the lack of thermal properties. In their response, the authors mentioned that they added further thermal conductivity data to improve their database. However, heat capacity and thermal diffusivity are still missing completely in this database. When reading the abstract and introduction etc. the additional thermal conductivity data is not mentioned. The term "thermal conductivity" does not appear in the text until section 3.6 "thermal properties" on page 26. To improve the structure of the article and for the sake of completeness, I suggest mentioning thermal conductivity already in the abstract and beginning of the article (e.g., in section 2 p.4 l. 95 you list all the petrophysical properties but not thermal conductivity and the mechanical properties). Furthermore, I recommend also evaluating the thermal conductivity and rock mechanical data and including the data in tables 4 and 5 as well as in the results section (thermal conductivity). I understand that the focus of this database lies on hydraulic properties. However, I strongly disagree with some of the author's responses stating that thermal properties are less important, in particular for geothermal assessments. Thus, the authors should clearly address the limitations of their work here. By e.g., adding the respective information to table 4, the reader is able to see which parameters can be correlated and which data was obtained on a separate sample set. Furthermore, the range of thermal conductivity of 1 W m-1K-1 is not small as stated by the authors ("1.5 to 2.5 W m-1K-1"). When looking at the original data in the database the data even ranges between 0.97 to 2.77 W m 1K 1. These rather large differences within one lithology should be addressed in combination with porosity/permeability and rock type/alteration in the results section.

The term "thermal conductivity" has been added to the abstract and the line in question in section 2. The number of samples analyzed for thermal conductivity as well as mechanical (strength) and acoustic properties has been added to Table 4. However, we choose not to include the averages for these properties into Table 5, as this table is focused on the best characterized properties (porosity, grain density, and intrinsic permeability). As noted in the manuscript, 52 out of the total 54 data points for thermal conductivity originate from samples obtained from a single lava flow, which shows a variability in thermal conductivity of ~1-2 W m$^{-1}$ K$^{-1}$. The other two data points originate from downhole core samples, which show higher thermal conductivity (~2.5-2.8 W m$^{-1}$ K$^{-1}$).

Due to the limited amount of data for thermal conductivity, we believe this data is insufficient to quantify how alteration and lithology control thermal conductivity. However, we have added the following text and figure to the results section on lines 704-716:

> Previous studies of Hawaiian basalts have shown that thermal conductivity decreases with increasing porosity and increases if the samples are saturated with water (Robertson and Peck, 1974). Although the thermal conductivity data in this study is mainly limited to samples derived a single unaltered lava flow in the Reykjavik area (Guðlaugsson, 2000), the

data suggest a similar relationship (Figure 16). Thermal conductivity measured at unsaturated conditions ranges from ~1-2 W m$^{-1}$ K$^{-1}$, with a general trend suggesting increasing thermal conductivity at lower connected porosity. In contrast, thermal conductivity measured under saturated conditions on two hyaloclastite samples obtained from the ÖJ-1 borehole is significantly higher, ranging from 2.5-2.75 W m$^{-1}$ K$^{-1}$. Although at present there is insufficient data to characterize the effect of lithology and alteration zone on thermal conductivity, Figure 16 indicates that significant variability in thermal conductivity within a single lithological unit results from the heterogenous distribution of pore space.

[Figure]

**Figure 16. Thermal conductivity as a function of connected porosity. Note that most of the available data is derived from a single unaltered lava flow (Guðlaugsson, 2000). Measurements on the lava flows were performed at unsaturated conditions, whereas measurements on the hyaloclastite samples were performed under water-saturated conditions.**

Figure 13a shows that compressional (P-wave) velocities are inversely correlated to porosity: basaltic intrusions show the highest velocities and the lowest porosities, while hyaloclastites have the lowest velocities and higher porosities. This relationship has been seen in several previous studies of volcanic rocks (e.g., Pola et al., 2014; Frolova et al., 2014; Wyering et al., 2014; Heap et al., 2015; Durán et al., 2019; Frolova et al., 2021).

Lines 684-691:

Alteration impacts rock strength and thereby exerts an influence on rock mechanical behavior and failure mode (e.g., Pola et al., 2014; Heap and Violay, 2021). Depending on the porosity changes during hydrothermal alteration and the abundance and type of clay minerals, alteration can increase or decrease rock strength (e.g., Wyering et al., 2014; Frolova et al., 2014; Pola et al., 2014; Mordensky et al., 2018; Farquharson et al., 2019; Heap et al., 2020a; Frolova et al., 2021). Figure 14 shows that uniaxial compressive strength (UCS) (Fig. 14a) and Young's modulus (Fig. 14b) decrease with increasing porosity, as has been observed in several previous studies of volcanic rocks (e.g., Al-Harthi et al., 1999; Pola et al., 2014; Wyering et al., 2014; Heap et al., 2014; Schaefer et al., 2015; Mordensky et al., 2018; Coats et al., 2018; Harnett et al., 2019). However, also consistent with these studies, the data reveal significant scatter; for example, at a porosity of 0.2, UCS can range from ~10 MPa to ~100 MPa. Heap and Violay (2021) describe how such variability in rock strength can result from variable hydrothermal alteration and the partitioning of porosity between pores and microcracks and their geometrical properties.

Thus, we have sought to emphasize in the text where observations derived from this dataset have also been seen in previous studies. In the interest of limiting the amount of text, we have sought to restrict the comparison to a general, rather than site-specific, level. However, in the revised version of the manuscript, we have added the following sentence to the Concluding remarks section (lines 749-753):

[revised manuscript text omitted]

• I disagree with some of the author's responses regarding the structure of the database and citing original data. The authors used another database published here in ESSD as a template to create their own database. Thus, a comment stating "modified from source XX" or "following the example of database XY" should be included in section 2. It clarifies on which basis this database was developed and acknowledges the original source and ideas.

There are several reasons why we believe that the link between this database and (presumably) Bär et al. (2020) is not as strong as the reviewer is suggesting:

- We had been working for more than a year on updating the previous version of the Valgarður database prior to ever seeing any files associated with Bär et al. (2020).
- We did not copy any data from Bär et al. (2020) directly into our database file; thus, it is not accurate to say that the database was "modified" from Bär et al. (2020). Although the two databases overlap, in both cases data was obtained from the primary sources (Gudmundsson et al., 1995; Franzson et al., 2011).

In addition, there are substantive differences in the structure of the two databases:

- In Bär et al. (2020), each row corresponds to an individual measurement on a sample; in our database, each row corresponds to a unique sample.
- The numerical petrographic classification scheme of Bär et al. (2020) is not incorporated in this database
- A description of alteration zone is not provided for most of the Iceland samples presented in Bär et al. (2020)
- Measurement conditions are not listed in separate columns as is the case for Bär et al. (2020), as the data presented in this database is collected at near-ambient pressure and temperature.
- Columns for the standard deviation, maximum and minimum value, number of measurements are not provided for each property

In the revised manuscript, we explicitly state the aspects of the Bär et al. (2020) database that served as a template for our database. We have added the following sentence to section 2 (lines 100-103) accordingly:

> "Following the example of Bär et al. (2020), we provide information about how each measurement was acquired in a 'Remarks' column adjacent to the reported value and set the fill colour of cells based on the type of data contained in the cell (e.g., cells listing the primary and secondary references are coloured yellow, cells containing sample meta-data are coloured blue, cells related to lithological characterization are coloured orange, etc.)."